# Microbiota-derived acetate enhances host antiviral response via NLRP3

Junling Niu [1,2], Mengmeng Cui[1], Xin Yang [2], Juan Li[1,3], Yuhui Yao[1,4], Qiuhong Guo[1], Ailing Lu[1], Xiaopeng Qi [5], Dongming Zhou[1], Chenhong Zhang [2] ✉, Liping Zhao [2,6] ✉ & Guangxun Meng [1,3,4] ✉

Pathogenic viral infections represent a major challenge to human health. Host immune responses to respiratory viruses are closely associated with microbiome and metabolism via the gut-lung axis. It has been known that host defense against influenza A virus (IAV) involves activation of the NLRP3 inflammasome, however, mechanisms behind the protective function of NLRP3 are not fully known. Here we show that an isolated bacterial strain, *Bifidobacterium pseudolongum* NjM1, enriched in the gut microbiota of *Nlrp3*[−/−] mice, protects wild-type but not *Nlrp3* deficient mice against IAV infection. This effect depends on the enhanced production of type I interferon (IFN-I) mediated by NjM1-derived acetate. Application of exogenous acetate reproduces the protective effect of NjM1. Mechanistically, NLRP3 bridges GPR43 and MAVS, and promotes the oligomerization and signalling of MAVS; while acetate enhances MAVS aggregation upon GPR43 engagement, leading to elevated IFN-I production. Thus, our data support a model of NLRP3 mediating enhanced induction of IFN-I via acetate-producing bacterium and suggest that the acetate-GPR43-NLRP3-MAVS-IFN-I signalling axis is a potential therapeutic target against respiratory viral infections.

Respiratory viral infection such as COVID-19 represents a major challenge to human health. Epidemics and pandemics caused by influenza A virus (IAV) infection are also major global threats. During IAV infection, the surface glycoprotein haemagglutinins (HA) of virions bind to host receptors first, then endocytosis mediates virus entry[1,2]. After entry, viral ribonucleoproteins (vRNP) and other proteins are synthesized and assembled into progeny virions to release and spread in the lung[3], which may cause lethality resulting from compromised virus clearance or deleterious immunopathology[4].

Epithelial cells in the respiratory epithelium are the main target cells for IAV replication[5], while innate immune cells such as macrophages and neutrophils are key components for sensing IAV infection and inhibiting virus early after infection[6]. In macrophages, multiple pattern recognition receptors (PRR) are engaged to initiate innate immune responses[7]. These include Toll-like receptor 7 (TLR7) sensing the viral single-stranded RNA (ssRNA) in the endosome[6,8], Toll-like receptor 3 (TLR3) sensing the viral double-stranded RNA (dsRNA) in the phagosome[6,9], and RIG-I-MAVS (retionoic acid-inducible gene I-mitochondrial antiviral signaling protein) sensing 5′-triphosphated viral RNA in the cytosol[6]. Recognition of IAV by these PRRs initiates the production of type I IFN (IFN-I) through the phosphorylation of IFN regulatory factors 3 and 7 (IRF3 and IRF7)[6]. Through interferon

[1]The Center for Microbes, Development and Health, CAS Key Laboratory of Molecular Virology & Immunology, Institut Pasteur of Shanghai, University of Chinese Academy of Sciences, 200031 Shanghai, China. [2]State Key Laboratory of Microbial Metabolism, School of Life Sciences and Biotechnology, Shanghai Jiao Tong University, 200240 Shanghai, China. [3]Nanjing Advanced Academy of Life and Health, 211135 Nanjing, China. [4]Pasteurien College, Soochow University, 215006 Suzhou, Jiangsu, China. [5]CAS Key Laboratory of Animal Models and Human Disease Mechanisms/Key Laboratory of Bioactive Peptides of Yunnan Province, Kunming Institute of Zoology, Chinese Academy of Sciences, 650201 Kunming, Yunnan, China. [6]Department of Biochemistry and Microbiology and New Jersey Institute for Food, Nutrition, and Health, School of Environmental and Biological Sciences, Rutgers University, NJ 08901, USA. ✉e-mail: zhangchenhong@sjtu.edu.cn; liping.zhao@rutgers.edu; gxmeng@ips.ac.cn

receptor, IFN-I induces multiple interferon-stimulated genes (ISG) to control virus replication[10], which is the most fundamental mechanism for viral containment by mammalian hosts.

Of note, anti-viral IFN-I response can be attributed to gut microbiota. Recent work showed that microbiota-induced IFN-I production by plasmacytoid dendritic cells (pDC) controls a distinct transcriptional, epigenetic, and metabolic basal state of conventional dendritic cells (cDC) to enhance anti-pathogen immune responses[11]. Outer membrane (OM)-associated lipooligosaccharide (LOS) of *Bacteroides fragilis* induces expression of IFN-β via TLR4-TRIF signaling in colonic cDCs to protect against vesicular stomatitis virus (VSV) and IAV infections[12]. In addition, the microbial metabolite desaminotyrosine (DAT), produced by *Clostridium orbiscindens*, also protects against IAV infection through the augmentation of IFN-I signaling[13]. In addition to IFN-I induction, microbiota-derived butyrate dampens deleterious immunopathology during IAV infection[4].

Both IFN-I dependent virus clearance and adequate control of immunopathology are necessary to protect against IAV infection. As such, the role of NLRP3 (NOD-, LRR- and pyrin domain-containing 3) in host response to IAV infection is intriguing. As the most studied inflammasome sensor protein, the major role of NLRP3 is to assemble ASC (apoptosis-associated speck-like protein containing a CARD) and caspase-1 to form inflammasome complex, which results in caspase-1 activation[14]. Activated caspase-1 cleaves pro-interleukin-1β (pro-IL-1β), pro-interleukin-18 (pro-IL-18), as well as Gasdermin-D, leading to the secretion of mature IL-1β/IL-18 and pyroptosis[15]. A number of studies showed that deficiency of NLRP3 led to more severe disease[16,17], and rescued or enhanced NLRP3 activity resulted in resistance to IAV infection[18,19]. These results indicate that NLRP3 is essential for the host to defeat IAV infection, however, the mechanisms behind remain largely elusive. Especially, whether the protective effect from NLRP3 involves gut microbiota-dependent type I IFN production has not been studied.

Here, we investigate the mechanisms associated with the role of NLRP3 in mediating gut microbiota-derived acetate-enhanced induction of IFN-I. We show that acetate enhances viral RNA-induced MAVS aggregation, IRF3 phosphorylation and type I IFN production; wherein NLRP3 mediates acetate-enhanced induction of IFN-I by interacting with the acetate receptor GPR43 (G-protein coupled receptor 43) and MAVS. Our findings indicate that microbiota-derived acetate enhances host antiviral response via NLRP3 thus protection against IAV infection, which may provide a base for designing intervention strategies against respiratory viral infections.

## Results

### Commensal microbiota from *Nlrp3*−/− mice enhances host defense against influenza A virus infection in WT animals

To test the possibility that NLRP3-dependent host defense against IAV infection may involve commensal microbiota, we co-housed *Nlrp3*−/− and WT mice at a 1:1 ratio from weaning to adulthood to exchange their microbiota, and then infected these mice with IAV. Consistent with previous reports, we found that influenza A virus (IAV) PR8 (H1N1) strain induced much more severe disease in *Nlrp3*−/− mice than their wild-type (WT) companions (Fig. 1a). Unexpectedly, Co-housed WT (Co-WT) mice showed enhanced defense against influenza virus infection compared with singly housed WT (Single-WT) mice, while Co-housed *Nlrp3*−/− mice (Co-*Nlrp3*−/−) showed unchanged susceptibility (Fig. 1a). This suggested that WT mice obtained IAV-resistant bacteria from *Nlrp3*−/− mice after co-housing, which failed to function in Co- and Single-*Nlrp3*−/− mice due to the absence of *Nlrp3* gene.

To test whether the augmented defense of Co-WT mice was mediated by commensal bacteria, those four groups of mice were given antibiotics in drinking water during co-housing before IAV infection. As expected, the difference between Co-WT and Single-WT

mice disappeared (Fig. 1b), which indicated that it was the special IAV-resistant bacteria that WT mice obtained from *Nlrp3*−/− mice during co-housing that enhanced host defense against IAV infection.

We then sequenced the 16S rRNA gene V3−V4 regions of the fecal bacteria and found that the overall structure of the gut microbiota showed a striking difference between Single-WT and Single-*Nlrp3*−/− mice, while it was converged between Co-WT and Co-*Nlrp3*−/− mice after co-housing (Fig. 1c). Next, we set out to identify the specific bacteria that enhanced Co-WT defense against IAV infection. First, we identified the amplicon sequence variants (ASV) that were significantly altered between Single-*Nlrp3*−/− mice and Single-WT mice using the algorithm LDA Effect Size (LEfSe) v1.0[20]. Notably, fifty-one ASVs were depleted in single-*Nlrp3*−/− mice, but twenty-five ASVs were enriched, especially ASV4 in the family Bifidobacteriaceae and ASV1 in the family Muribaculaceae (Fig. 1d). Then, we found that nineteen ASVs including ASV4 and ASV1 were enriched in Co-WT mice after co-housing with Co-*Nlrp3*−/− mice, while six ASVs were depleted in Co-WT mice (Fig. 1e). At last, we identified seven ASVs that were enriched and three ASVs that were depleted in Co-WT mice due to co-housing with Co-*Nlrp3*−/− mice, especially ASV4 and ASV1 (Fig. 1f, g).

To obtain their pure cultures, we performed sequence-guided screening of bacterium colonies from fresh feces of singly housed *Nlrp3*−/− mice. A pure culture was obtained using a previously reported mGAM medium (Nissui 05426). We named this strain as NjM1, which contains 4 copies of the 16S rRNA gene in its genome and had 100% identity with ASV4, which were enriched in Co-WT mice but not Single-WT mice (Fig. 1g, h and Supplementary Fig. 1a, b). According to whole 16S rRNA gene sequence, the nearest neighbor of NjM1 in GenBank is *Bifidobacterium pseudolongum subsp. globosum* strain RU224 with 97.96−98.42% homology (Fig. 1h), indicating that the strain NjM1 is belonged in *Bifidobacterium pseudolongum*. *B. pseudolongum* NjM1 grows fast to plateau stage 8 hours after inoculation under anaerobic culture condition (Supplementary Fig. 1c), and bacteria cultured under this condition was applied for functional experiments. Taken together, microbiota from *Nlrp3*−/− mice enhances host defense against influenza A virus infection in WT animals, and the gut microbiota of *Nlrp3*−/− mice contains abundant *B. pseudolongum* NjM1, which is associated enhanced protection against IAV in Co-WT but not *Nlrp3*−/− mice.

### Elevated levels of microbial metabolite acetate associates with *B. pseudolongum* NjM1 that produces acetic acid

To find out metabolites that may have been increased in WT mice after co-housing with *Nlrp3*−/− mice which possibly enhanced defense against IAV infection, orthogonal partial least-squares discriminant analysis (OPLS-DA) was carried out for ¹H nuclear magnetic resonance (¹H NMR) spectral data obtained from feces samples. The results suggested that metabolomes of feces from Co-WT mice differed markedly from those of Single-WT mice (Fig. 2a, Supplementary Data 1). Co-housing with *Nlrp3*−/− mice caused marked elevations of the acetate and butyrate levels and reduction in trimethylamine (TMA) in the feces of Co-WT mice (Fig. 2b, c, Supplementary Data 1). High level of acetate was also detected in the feces of Single-*Nlrp3*−/− mice (Supplementary Fig. 2a-c, Supplementary Data 1). On the other hand, we detected acetate in circulation using peripheral blood and confirmed that acetate in Co-housed WT mice with *Nlrp3*−/− mice was higher than that in Single-WT mice (Supplementary Fig. 2d). Notably, the isolated *B. pseudolongum* NjM1 mentioned above, a predominant bacterium in Co-WT mice, was able to produce acetic acid specifically (Fig. 2d), which was closely related to the significantly elevated acetate levels in the feces and circulation of Co-WT mice. These data indicate that *B. pseudolongum* NjM1 isolated from the gut microbiota of *Nlrp3*−/− mice may produce acetate in vivo to confer host protection against IAV infection in Co-WT but not *Nlrp3*−/− animals.

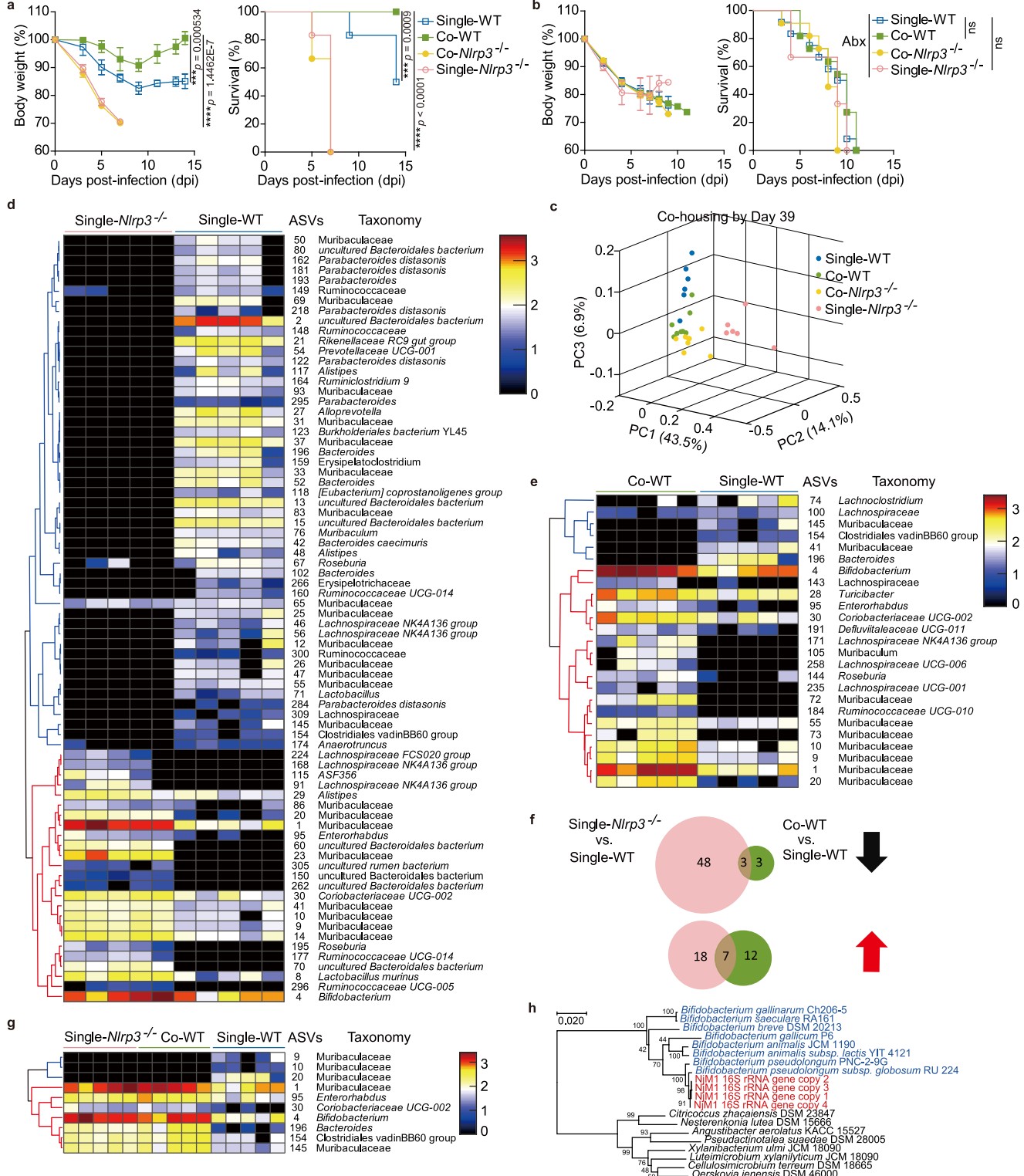

## *B. pseudolongum* NjM1 and acetate protect WT but not *Nlrp3*⁻/⁻ mice against IAV infection

We next wondered whether *B. pseudolongum* NjM1 or its metabolite acetate was able to confer protection against influenza virus challenge. To test these hypotheses, we first inoculated WT mice with *B. pseudolongum* NjM1 by oral gavage, or with NjM1 conditioned supernatant by intranasal administration before IAV infection. Of note, inoculation of *B. pseudolongum* NjM1 bacterium or conditioned supernatant both dramatically decreased IAV-induced weight loss and mortality (Fig. 3a, b, Supplementary Fig. 3a). However, the same inoculation of NjM1 and

conditioned supernatant didn't rescue IAV-induced weight loss and mortality in *Nlrp3*⁻/⁻ mice (Fig. 3a, b, Supplementary Fig. 3a), suggesting that NLRP3 was required for *B. pseudolongum* NjM1-mediated protection against IAV infection, consistent with the susceptibility of *Nlrp3*⁻/⁻ mice to IAV infection even though they are rich in *B. pseudolongum* NjM1 in the gut (Fig. 1a, d).

To explore whether acetate itself was sufficient to confer protection against IAV infection, WT mice were given sodium acetate (SA) in drinking water 2 weeks in advance and during IAV infection. The results showed that acetate significantly reduced body weight loss

**Fig. 1 | Commensal microbiota from *Nlrp3*⁻ᐟ⁻ mice enhances host defense against influenza A virus infection in WT animals. a** *Nlrp3*⁻ᐟ⁻ and WT mice either singly housed or co-housed were intranasally infected with PR8. The body weights and survival rates were assessed. Single-WT (*n* = 12), Single-*Nlrp3*⁻ᐟ⁻ (*n* = 12), Co-WT (*n* = 18), Co-*Nlrp3*⁻ᐟ⁻ (*n* = 18). **b** All groups of mice as in (**a**) were given antibiotics, then infected and assessed as in (**a**). Single-WT (*n* = 12), Single-*Nlrp3*⁻ᐟ⁻ (*n* = 9), Co-WT (*n* = 11), Co-*Nlrp3*⁻ᐟ⁻ (*n* = 11). **c−g** Fecal microbiota composition was assayed for mice in (**a**) by bacterial 16S rRNA gene V3−V4 region sequencing before PR8 infection: (**c**) Principal coordinate analysis (PCoA) plot (based on Bray−Curtis distance) of the gut microbiota; (**d, e, g**) Heatmaps for ASVs that were significantly altered between these groups, as identified using the algorithm LDA Effect Size (LEfSe) v1.0 (*n* = 5 per group). The cluster tree on the left of each heatmap shows

associations among these ASVs, as determined by the Spearman correlation coefficient based on their relative abundances among all the samples. The heat maps show the relative abundance (log₁₀ transformed) of each ASV in a sample from an individual mouse; (**f**) Venn diagram of differential ASVs in relative abundance. **h** A phylogenetic tree constructed with the 16S rRNA genes of *B. pseudolongum* NjM1 (red), other bacteria within the family Bifidobacteriaceae (blue) and typical bacterial strains within the phylum Actinobacteria. Results represent three (**a**) or two independent experiments (**b**). Changes in body weights shown in (**a, b**) are presented as mean ± SEM, one-way ANOVA with Tukey's post-hoc test. Survival rates shown in (**a, b**) are analyzed with Log-rank (Mantel−Cox) test. Significant values are defined by *$p < 0.05$, **$p < 0.01$, ***$p < 0.001$, ****$p < 0.0001$. Source data are provided as a Source Data file.

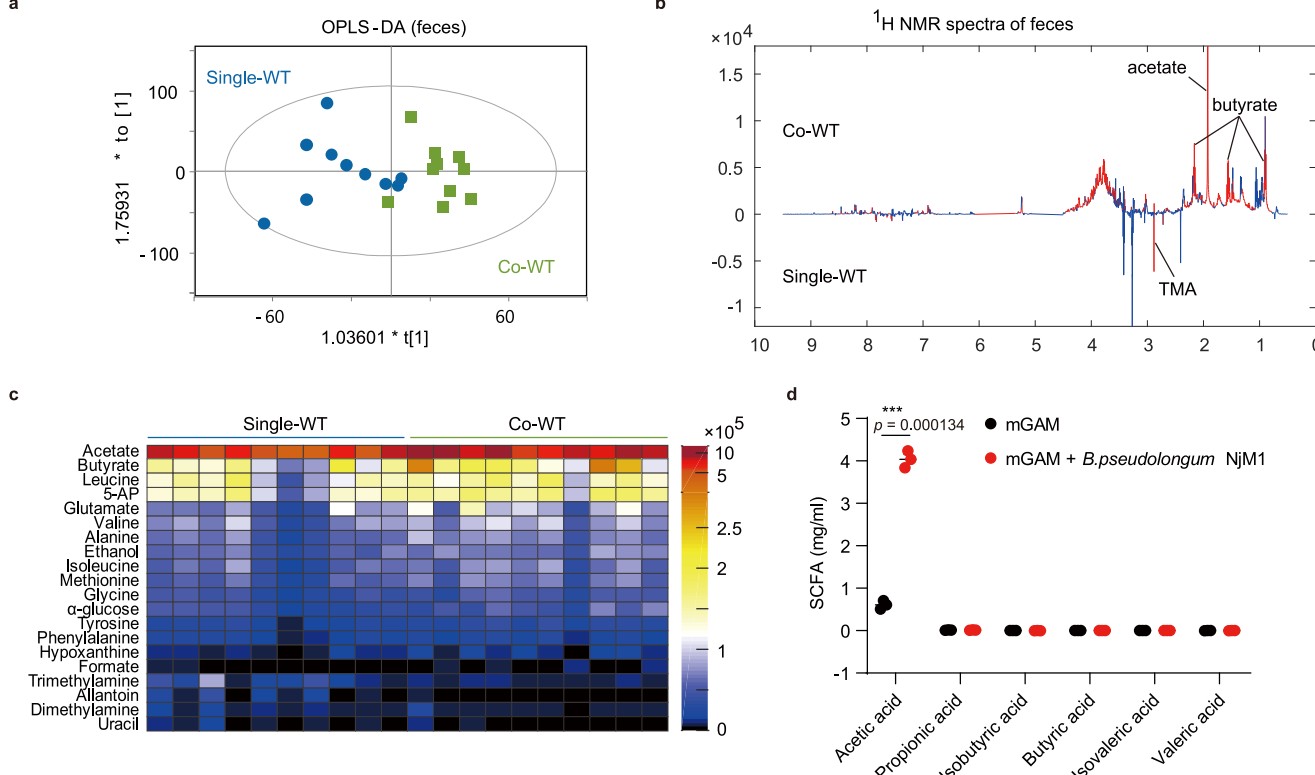

**Fig. 2 | Fecal concentration of acetate in Co-WT mice increases and *B. pseudolongum* NjM1 produces acetic acid. a−c** Cross-validated OPLS-DA scores plots (**a**) and the corresponding loadings plots (**b**) and heat map (**c**) of feces extracts data from Single-WT mice and Co-WT mice on day 39 after co-housing with *Nlrp3*⁻ᐟ⁻ mice. (*n* = 10 per group. OPLS-DA: $Q^2$ = 0.224. CV-ANOVA: $P$ = 0.041). 5-AP, aminopentanoate; TMA, trimethylamine. **d** The concentrations of SCFAs from *B.*

*pseudolongum* NjM1 conditioned supernatant were measured using gas chromatography, and the amounts are presented as milligram per milliliter of supernatant. Results represent three independent experiments (**d**). Data in (**d**) are presented as mean ± SEM, two-tailed Student's *t* test. Significant values are defined by *$p < 0.05$, **$p < 0.01$, ***$p < 0.001$, ****$p < 0.0001$. Source data are provided as a Source Data file.

caused by IAV infection and increased the survival rate in WT, but not *Nlrp3*⁻ᐟ⁻ mice (Fig. 3c), suggesting that treatment with acetate also had a NLRP3-dependent protective effect against IAV. To further assess whether acetate-releasing diet can protect against IAV infection, WT mice were fed specialized diets HAMS (high-amylose maize starch) or HAMSA (acetylated HAMS) that releases large amounts of acetate after bacterial fermentation in the gut as previously described[21–24]. Indeed, HAMSA-fed mice had higher concentration of acetate in cecal contents, feces and peripheral venous blood than that of HAMS-fed mice (Supplementary Fig. 3e). Of note, acetate-releasing HAMSA significantly reduced body weight loss caused by IAV infection in WT, but not *Nlrp3*⁻ᐟ⁻ mice (Fig. 3d), suggesting a diet that releases acetate also protected against IAV in a NLRP3-dependent manner. We also tried to treat mice with acetate in the drinking water simultaneously with IAV infection (Supplementary Fig. 3b), or inoculated mice with acetate by

intranasal route post-intranasal IAV instillation (Supplementary Fig. 3c). Notably, under these therapeutic protocols, acetate protected mice against weight loss and mortality caused by IAV infection as well (Supplementary Fig. 3b, c). When sodium acetate was added to the drinking water before infection and withdrawn after infection, lower weight loss was evident only at the early stage after IAV instillation, and the effect disappeared later on (Supplementary Fig. 3d). These data demonstrated that as long as the acetate was applied to the mice early enough without being withdrawn, the mice were able to be protected against IAV infection.

When IAV-infected mice were harvested on day 7 post-infection, pathology analysis and viral titration revealed that the reduced body weight loss and increased survival rate in acetate-treated mice associated with decreased inflammatory cell infiltration, as well as reduced viral load (Fig. 3e, f). However, this effect was evident only in the WT

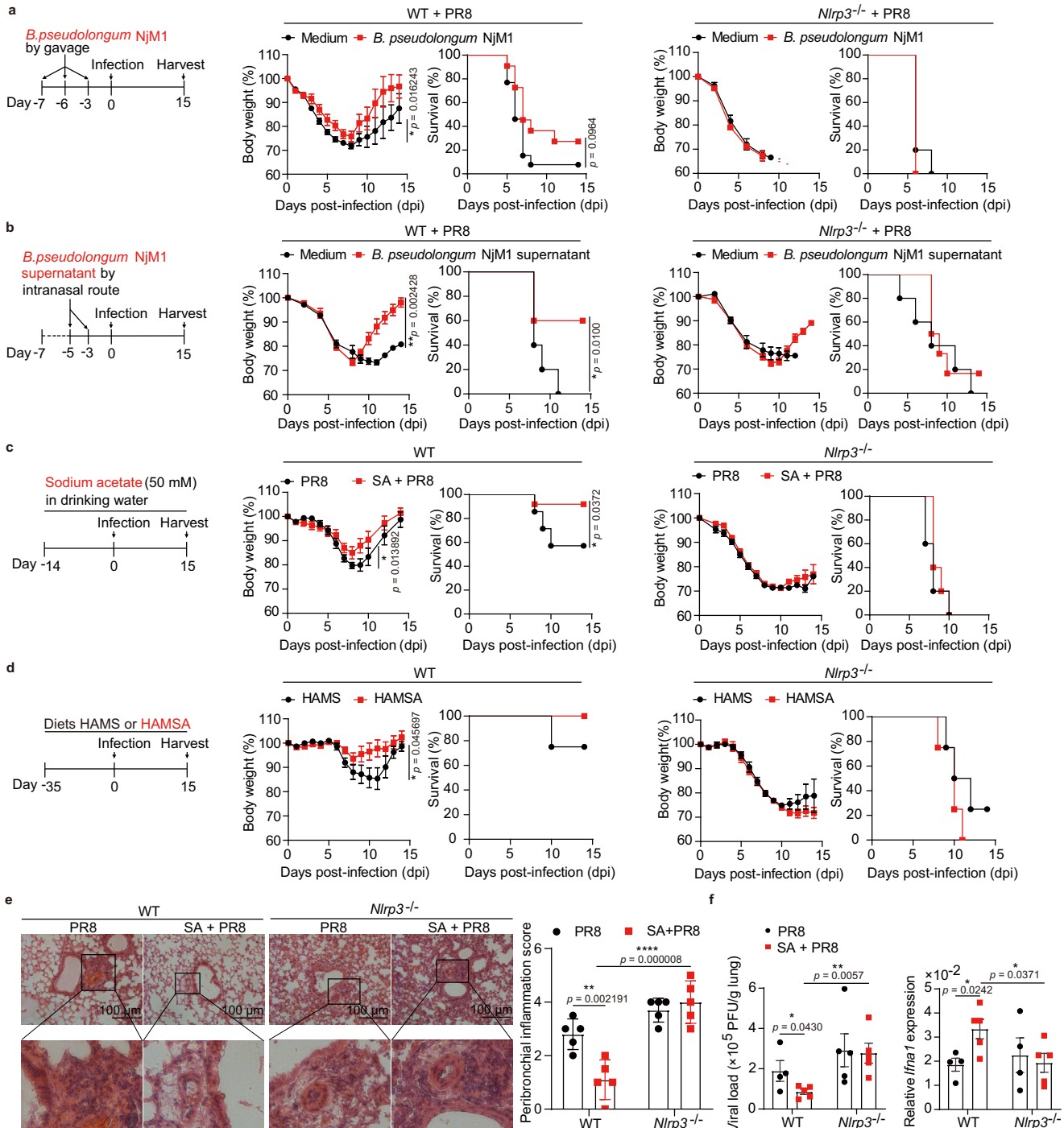

**Fig. 3 | *B. pseudolongum* NjM1 and acetate protect WT but not *Nlrp3*[−/−] mice against IAV infection. a–d** WT and *Nlrp3*[−/−] mice were given: (**a**) either $1.48 \times 10^9$ *B. pseudolongum* NjM1 or mGAM medium control for each mouse by gavage 3 times (WT, *n* = 11 or 13; *Nlrp3*[−/−], *n* = 10 or 10), (**b**) either 50 μL *B. pseudolongum* NjM1 conditioned supernatant or mGAM medium for each mouse by intranasal route twice (WT, *n* = 10 or 10; *Nlrp3*[−/−], *n* = 12 or 10), (**c**) drinking water with or without 50 mM sodium acetate (WT, *n* = 14 or 14; *Nlrp3*[−/−], *n* = 10 or 10), (**d**) either diet HAMS or HAMSA for 5 weeks (WT, *n* = 8 or 8; *Nlrp3*[−/−], *n* = 8 or 8), and then intranasally infected with influenza A virus PR8 (H1N1). Body weight changes in percentage and survival rates of such mice post-infection were assessed. **e** The lungs from infected WT and *Nlrp3*[−/−] mice treated as in (**c**) were

harvested on day 7 post-infection, sectioned, and stained with hematoxylin and eosin. Representative images and peribronchial inflammation scores are shown. **f** Viral loads and relative *Ifna1* expression in the lung homogenates from infected WT and *Nlrp3*[−/−] mice were determined with the standard plaque assay and qPCR, respectively, on day 7 post-infection. Results represent two (**a**, **b**, **d**–**f**) or three independent experiments (**c**). Changes in body weights shown in (**a**–**d**) are presented as mean ± SEM, two-tailed Student's *t* test. Survival rates shown in (**a**–**d**) are analyzed with Log-rank (Mantel−Cox) test. Data in (**e**–**f**) are presented as mean ± SEM, one-way ANOVA with Dunnett's post-hoc test. Significant values are defined by **p* < 0.05, ***p* < 0.01, ****p* < 0.001, *****p* < 0.0001. Source data are provided as a Source Data file.

mice, but not in mice deficient for *Nlrp3* (Fig. 3e, f). Of note, acetate-treated WT mice showed elevated levels of IFN-I in the lung, which was not observed in *Nlrp3*$^{-/-}$ mice (Fig. 3f). These results implied that acetate may have induced IFN-I production in a NLRP3-dependent manner to contain IAV in the lung.

## Acetate protects against IAV infection in an IFNAR1-dependent manner

To investigate whether acetate suppressed IAV replication through IFN-I signaling pathway, *Ifnar1*$^{-/-}$ mice were subjected to IAV infection after treatment with acetate in drinking water. Of note, the protective effect of acetate observed in WT mice was totally lost in such mice (Supplementary Fig. 4a). No matter treated with acetate or not, all the *Ifnar1*$^{-/-}$ mice lost body weight quickly and similarly, and died finally (Fig. 4a, Supplementary Fig. 4a). Pathological analysis and viral titration showed that both acetate-treated and -untreated mice exhibited obvious inflammatory cell infiltration around the bronchi and blood vessel, as well as high and comparable viral load in the lung (Fig. 4b, c). Although the induction of IFN-I was more or less normal in the *Ifnar1*$^{-/-}$ mice, interferon-stimulated genes (ISGs) such as *Isg15* or *Oas1a* were not induced due to the lack of type I interferon receptor (Supplementary Fig. 4b). These data thus demonstrated that acetate-mediated suppression of IAV infection in vivo was fulfilled through type I interferon signaling.

We next asked which type of host cells were involved in the protective effect of sodium acetate. To this end, we targeted the most abundant innate immune cells in the lung, which are alveolar macrophages[6]. Clodronate-mediated depletion of alveolar macrophages in the lung showed that once these cells were depleted, there was no difference in weight loss or survival between acetate-treated and the control group anymore (Fig. 4d and Supplementary Fig. 4c), suggesting that macrophages play a key role in acetate-enhanced defense against IAV infection.

## Acetate controls IAV replication by enhancing IFN-I induction via the MAVS-TBK1-IRF3 axis

The next question we asked was how acetate induces or helps with inducing IFN-I production. We first found that acetate-treated bone marrow-derived macrophages (BMDM) showed clearly reduced IAV replication at 24 h post-infection, as indicated by reduced viral nucleoprotein (NP) protein levels (Fig. 5a) and decreased viral M1 gene expression (Fig. 5b). In addition, the amount of infectious viral particles in the supernatant of BMDMs was also reduced (Fig. 5c). Notably, acetate treatment promoted *Ifna1* and *Ifnb1* expression in BMDMs at 6 h and 24 h after IAV infection (Fig. 5d). We confirmed that acetate inhibited virus replication through enhancing antiviral immune responses in macrophages, rather than by directly destroying the viral particles, because co-incubating IAV particles with 250 μM acetate (a non-toxic concentration determined by 7AAD staining) for various times in vitro did not affect the infectivity of influenza virus (Supplementary Fig. 5a, b). We also confirmed that sodium was not responsible for enhanced *Ifna1* expression and the inhibition of virus replication, as both sodium acetate (SA) and potassium acetate (PA) showed a similar effect, while sodium butyrate (SB) did not (Supplementary Fig. 5e).

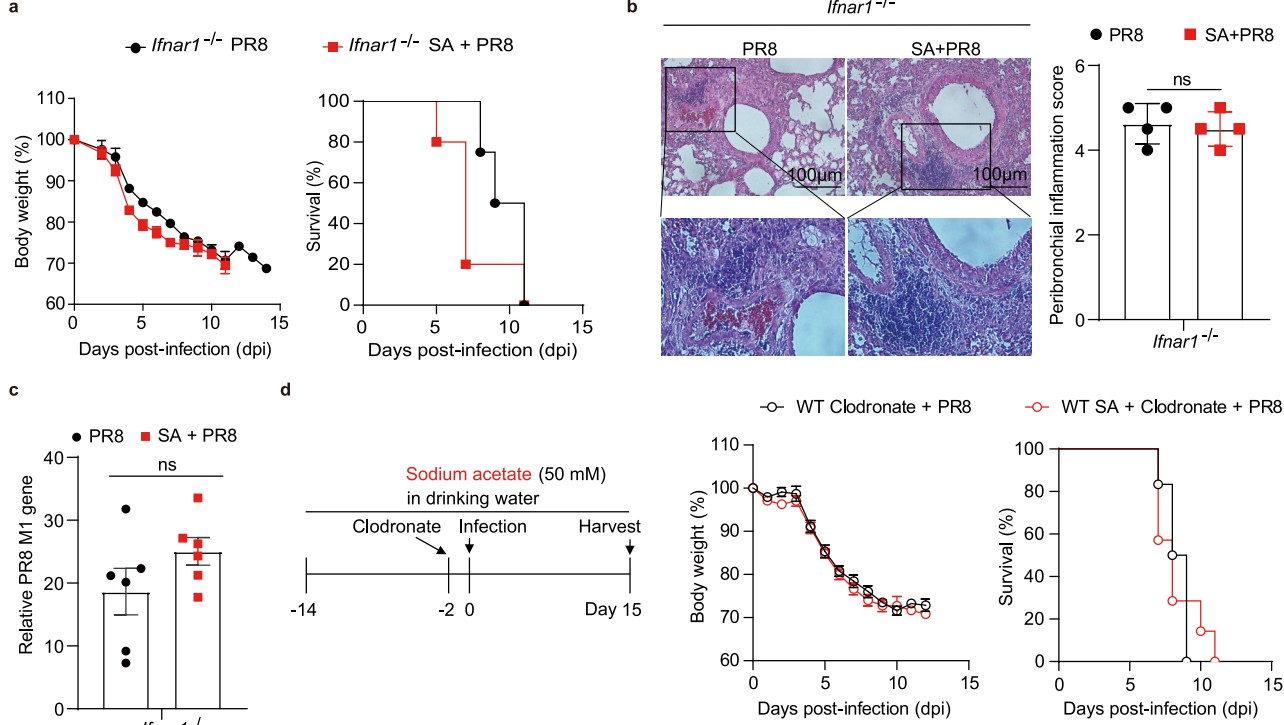

**Fig. 4 | Acetate protects against IAV infection in an IFNAR1-dependent manner.**
**a** *Ifnar1*$^{-/-}$ mice pretreated with sodium acetate (SA) ($n = 10$) or drinking water ($n = 8$) were intranasally infected with PR8 as depicted in Fig. 3c, weight loss and survival were monitored daily for 14 days, and percentage of body weight post-infection relative to initial body weight (day 0) and survival rate were presented. **b** *Ifnar1*$^{-/-}$ mice ($n = 4$, $n = 4$) were pretreated and infected as in (**a**), and the lungs were harvested on day 7 post-infection, sectioned, and stained with hematoxylin and eosin. Representative images and peribronchial inflammation scores are shown. **c** Relative PR8 *M1* gene expression to *Gapdh* in lung tissue of *Ifnar1*$^{-/-}$ mice ($n = 6$, $n = 6$) on day 7 post-infection. **d** WT mice were pretreated with sodium acetate (SA) ($n = 14$) or drinking water ($n = 12$) and then intranasally administered with clodronate to deplete macrophages in the lung. These two groups of mice were infected with PR8, weight loss and survival were monitored daily for 14 days, and percentage of body weight post-infection relative to initial body weight (day 0) and survival rate were presented. Results represent three (**a**–**c**) or two independent experiments (**d**). Data in (**a**, **d**) (body weight) and (**b**, **c**) are presented as mean ± SEM, two-tailed Student's *t* test. Survival rates shown in (**a**, **d**) are analyzed with Log-rank (Mantel–Cox) test. ns, not significant. Source data are provided as a Source Data file.

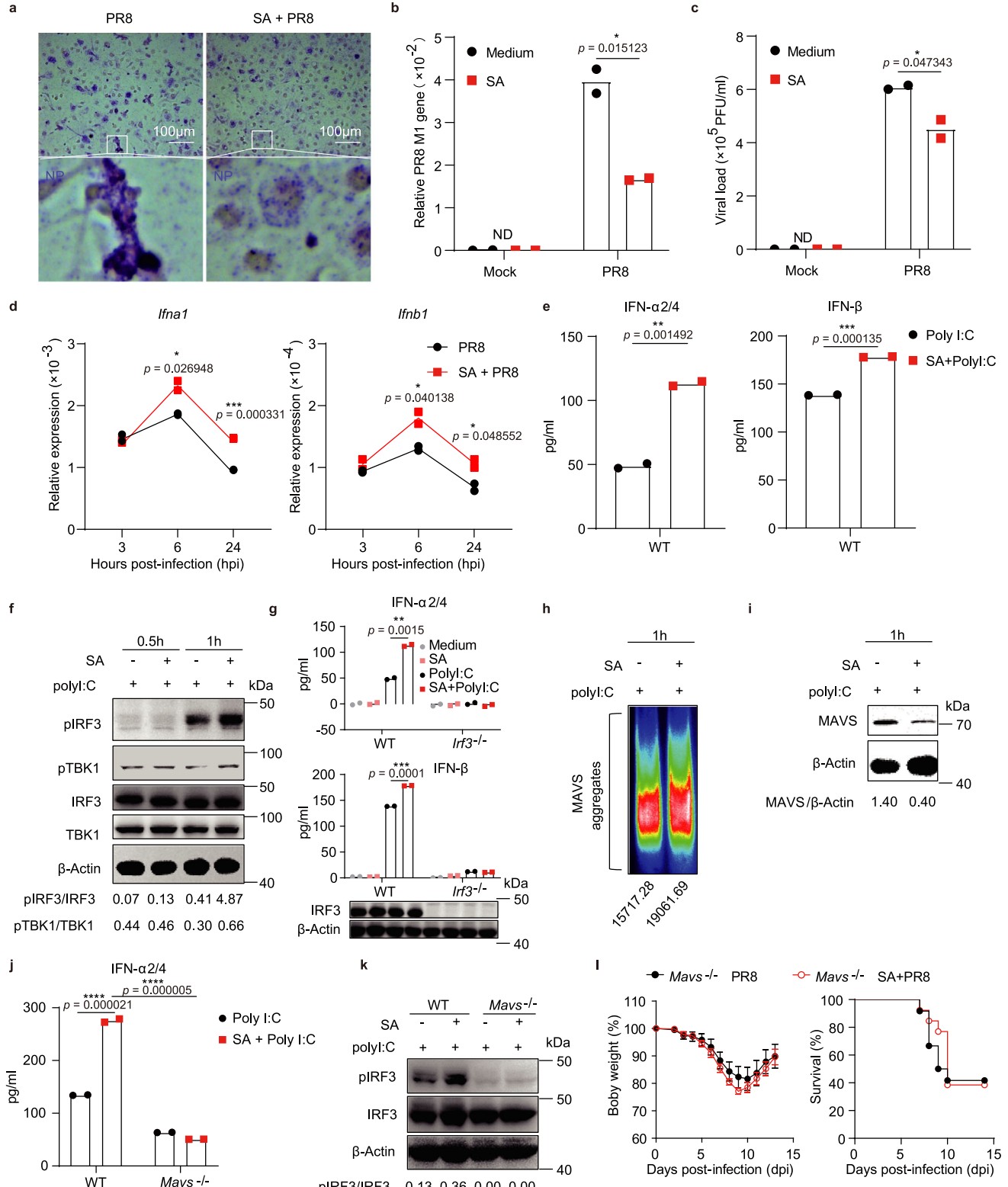

We further found that acetate treatment enhanced the ability of BMDMs to absorb and endocytose virus particles, as acetate-treated cells bound and endocytosed more virus particles, and fewer were left in the supernatant (Supplementary Fig. 5c, d). Therefore, acetate controlled IAV replication by enhancing IFN-I generation in macrophages.

To investigate the molecular mechanism on IFN-I production promoted by acetate, BMDMs were treated with acetate for 24 h and stimulated with polyinosinicpolycytidylic acid (polyI:C), a structural analog of double-stranded RNA which can functionally mimic IAV RNA to induce IFN-I production (Supplementary Fig. 5f). In such experiment, acetate promoted IFN-I production significantly (Fig. 5e and Supplementary Fig. 5g), for which sodium was not responsible, because both sodium acetate (SA) and potassium acetate (PA) showed similar effect, while sodium butyrate (SB) did not (Supplementary Fig. 5h). As IFN-I induction is largely dependent on interferon

**Fig. 5 | Acetate controls IAV replication by enhancing IFN-I induction via the MAVS-TBK1-IRF3 axis in macrophages. a–c** BMDMs were pretreated with 250 μM SA for 24 h and then infected with PR8 (MOI = 2) for 24 h. **a** Representative images of PR8-infected BMDMs after staining of viral nucleoprotein (NP, blue). **b** Relative PR8 *M1* gene expression to *Gapdh* in BMDMs (*n* = 2). **c** Viral load in the supernatant of infected BMDMs (*n* = 2). **d** BMDMs were pretreated with 250 μM SA for 24 h and then infected with PR8 for 3 h, 6 h or 24 h. Relative *Ifna1* and *Ifnb1* expression to *Gapdh* was determined (*n* = 2). **e–k** BMDMs were pretreated with 250 μM SA for 24 h and then transfected with 200 ng polyI:C for 12 h (**e, g, j**) or 0.5 h – 1 h (**f, h, i, k**): (**e, g, j**) Concentrations of IFN-α2/4 and IFN-β released from BMDMs (*n* = 2) were determined by ELISA; (**f, i, k**) The cell lysates were subjected to immunoblot analysis with indicated antibodies. The relative ratios of pIRF3 (Ser396) to total IRF3, pTBK1 (Ser172) to total TBK1 and MAVS monomer to β-Actin were analyzed with

ImageJ v2.0 and marked below; (**h**) Crude mitochondria extracts were prepared from the BMDMs and then subjected to semidenaturing detergent agarose gel electrophoresis (SDD-AGE) and immunoblotted with rabbit anti-MAVS antibody, and the intensity of MAVS aggregation was analyzed with ImageJ v2.0 and marked below. **l** *Mavs*[−/−] mice were given drinking water with or without 50 mM SA (*n* = 13 or 12), and then intranasally infected with PR8. Body weights and survival rates were assessed. Results represent three (**a–i**) or two independent experiments (**j–l**). Data in (**b–e, g, j, l**) (changes of body weight) are presented as mean (± SEM), two-tailed Student's *t* test (**b–e, g, l**), one-way ANOVA with Dunnett's post-hoc test (**j**). Survival rates shown in (**l**) are analyzed with Log-rank (Mantel–Cox) test. Significant values are defined by *$p < 0.05$, **$p < 0.01$, ***$p < 0.001$, ****$p < 0.0001$. Source data are provided as a Source Data file.

regulatory factors (IRF), we thus checked the expression and post-translational modifications of the dominant transcriptional factor IRF3 in viral RNA-induction of IFN-I. Here we found that acetate treatment enhanced IRF3 phosphorylation (pIRF3) (Fig. 5f). As expected, TANK binding kinase 1 (TBK1) phosphorylation (pTBK1) in these cells were also elevated upon acetate treatment (Fig. 5f), while no significant difference in IRF7 phosphorylation between acetate-treated and control BMDMs was notified (Supplementary Fig. 5i). Accordingly, acetate was not able to enhance any IFN-I production in *Irf3*[−/−] BMDMs (Fig. 5g).

In response to viral infection, mitochondrial antiviral signaling (MAVS) protein monomers form functional aggregates, which activate the transcription factor IRF3 to induce type I interferons[25]. Therefore, we believed that acetate had promoted polyI:C-induced MAVS aggregation to enhance IRF3 phosphorylation (pIRF3). This turned out to be true as indicated by semidenaturing detergent agarose gel electrophoresis (SDD-AGE) results: acetate-treated and polyI:C-stimulated BMDMs exhibited clearly more MAVS aggregates on mitochondria compared with polyI:C alone stimulated BMDMs (Fig. 5h). At the same time, less MAVS protein in the monomer form was detected in cell lysate-derived supernatant from acetate-treated and polyI:C-stimulated BMDMs compared to polyI:C alone stimulated cells (Fig. 5i).

As expected, acetate lost the capacity to enhance IFN-I production and IRF3 phosphorylation in *Mavs*[−/−] BMDM (Fig. 5j, k), nor to protect against IAV infection in *Mavs*[−/−] mice (Fig. 5l), demonstrating that MAVS was indeed involved in the mechanism by which acetate protected the mice from viral infection.

## Acetate promotes MAVS aggregation through GPR43

There are specific and shared cellular receptors for short-chain fatty acids (SCFAs): propionate is an agonist for both G-protein coupled receptor 41 (GPR41) and G-protein coupled receptor 43 (GPR43), butyrate is more active on GPR41, whereas acetate is more selective for GPR43[26]. To investigate whether acetate promoted IFN-I production through GPR43, we first analyzed the bronchoalveolar lavage fluid (BALF) from IAV-infected WT mice, and found reduced viral load and increased concentrations of IFN-α and IFN-β in the BALF of acetate-treated mice (Fig. 6a, b). What's more, sodium acetate (SA) treatment in these mice upregulated the expression of GPR43 on CD11b[+]F4/80[+] macrophages in the BALF (Fig. 6c, Supplementary Fig. 6a). Next, we investigated the potential role of GPR43 for acetate-enhanced IFN-I induction. It was found that BMDMs expressed GPR43 (Supplementary Fig. 6b), and GPR43 protein level was increased by acetate treatment (Supplementary Fig. 6c). Importantly, GPR43 was activated directly by acetate, as indicated by increased intracellular Ca[2+] mobilization[27,28] in WT but not *Gpr43*[−/−] BMDMs (Supplementary Fig. 7a, b). Interestingly, 4-CMTB, a GPR43 agonist, also augmented polyI:C-induced IFN-α production from BMDMs (Fig. 6d), indicating an important role of GPR43 in promoting IFN-I production. Of note, *Gpr43* deficiency or silencing with siRNA or GPR43 blocking with neutralizing antibody all compromised acetate function in promoting IFN-α production

(Fig. 6e, Supplementary Fig. 6d, e, g). Accordingly, acetate-induced MAVS aggregation was clearly compromised when GPR43 was knockout or silenced with siRNA (Fig. 6f, Supplementary Fig. 6f). The protective role of acetate against IAV infection was also compromised in *Gpr43*[−/−] mice (Supplementary Fig. 6h). Thus, acetate promoted polyI:C-induced MAVS aggregation and IFN-I production through GPR43 receptor in macrophages.

To test the possibility of GPR43 interaction with MAVS, HEK293T cells were transfected with a plasmid expressing V5-tagged GPR43, along with a plasmid expressing FLAG-tagged MAVS. Co-Immunoprecipitation (Co-IP) and Western blot analysis demonstrated the interaction of MAVS with GPR43 (Fig. 6g). Moreover, fluorescence images demonstrated co-localization of MAVS with GPR43 in acetate-treated BMDMs upon polyI:C stimulation (Fig. 6h). Together, these data suggested that acetate treatment promoted GPR43 interaction with MAVS in BMDMs after polyI:C stimulation, which promotes subsequent MAVS aggregation and IFN-I induction.

## NLRP3 participates in acetate-promoted IFN-I generation via aiding MAVS aggregation

In the beginning of our study, we found that WT mice co-housed with *Nlrp3*[−/−] mice (Co-WT) showed enhanced defense against IAV infection (Fig. 1a), which has been demonstrated to be resulted from increased *B. pseudonlongum* NjM1 producing acetic acid, while *Nlrp3*[−/−] mice with comparable NjM1 were susceptible to IAV infection (Fig. 1a, g). Based on the observations mentioned above, we hypothesized that NLRP3 might have been involved in the acetate-enhanced IFN-α production during IAV infection. To test this possibility, we first performed in vivo experiments in which WT and *Nlrp3*[−/−] mice were treated with acetate and infected with IAV. Here we found that *Nlrp3*[−/−] mice showed defective expression of *Ifna1* (Fig. 7a), and bore significantly higher titer of virus in the lung compared to WT mice 3 days after infection (Fig. 7b). Accordingly, *Nlrp3*[−/−] mice exhibited significantly more weight loss compared with WT mice (Fig. 7c). In addition, we performed in vitro experiments, wherein WT and *Nlrp3*[−/−] BMDMs were treated with acetate and then stimulated with polyI:C. Notably, acetate promoted IFN-α production in WT BMDMs, but *Nlrp3* deficiency impaired such function (Fig. 7d and Supplementary Fig. 8a). The results could not be attributed to cell differences due to lack of *Nlrp3* expression, because there was no difference in CD11b, F4/80 or GPR43 expression between *Nlrp3*[−/−] and WT BMDMs (Supplementary Fig. 8b, c). The GPR43 intensity displayed in *Nlrp3*[−/−] BMDMs was also comparable to that in WT BMDMs after acetate treatment plus polyI:C stimulation (Supplementary Fig. 8d). A study from Macia, L. et al[29]. reported that the SCFA acetate binds to GPR43 on colonic epithelial cell lines to stimulate K[+] efflux and hyperpolarization, leading to NLRP3 inflammasome activation, which is essential for gut homeostasis[29]. But acetate alone or acetate plus Poly I:C did not induce IL-1β release from macrophages (Supplementary Fig. 8a), indicating that acetate functions differently when it acts on different tissues or on different cell types.

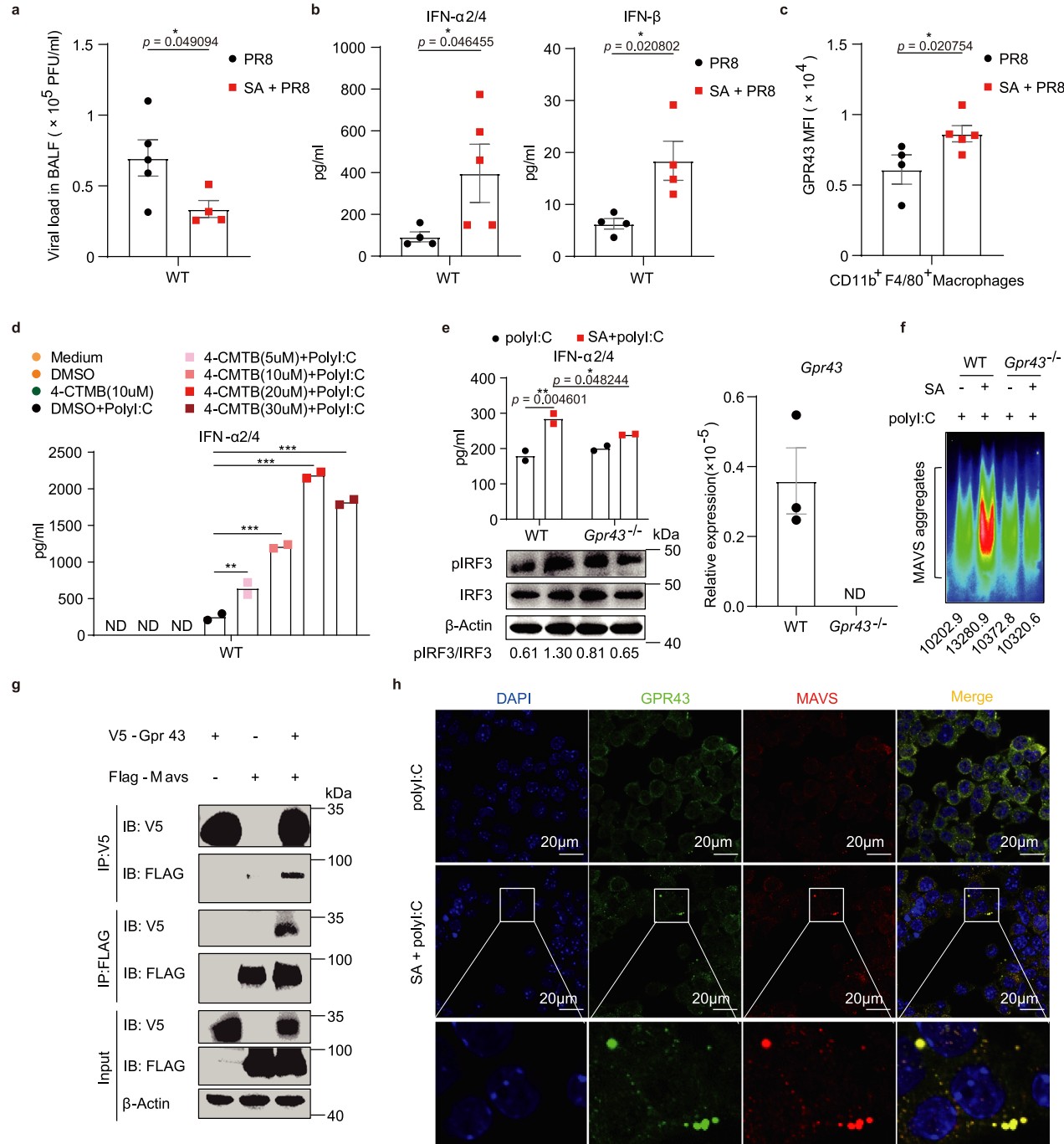

**Fig. 6 | Acetate promotes MAVS aggregation through GPR43. a–c** WT mice were given drinking water with or without 50 mM SA and then intranasally infected with PR8. **a** Viral load in the BALF from infected WT mice ($n = 5$, $n = 4$) on day 3 post-infection. **b** Concentrations of IFN-α2/4 and IFN-β in the BALF from infected WT mice ($n = 4$, $n = 5$; $n = 4$, $n = 4$) on day 3 post-infection. **c** Mean fluorescence intensity (MFI) of GPR43 expressed on CD11b[+]F4/80[+] macrophages in the BALF from infected WT mice ($n = 4$, $n = 5$) on day 3 post-infection. **d** BMDMs were pretreated with DMSO or 4-CMTB for 24 h and then stimulated and analyzed as in Fig. 5e ($n = 2$). **e** WT or *Gpr43*[−/−] BMDMs were pretreated with 250 μM SA for 24 h and then transfected with 200 ng polyI:C for 12 h (upper) ($n = 2$) or 1 h (bottom). Concentrations of IFN-α2/4 released from BMDMs were determined by ELISA. The cell lysates were subjected to immunoblot analysis. The relative ratios of pIRF3 (Ser396) to total IRF3 were analyzed with ImageJ v2.0 and marked below. *Gpr43* mRNA expression was determined by qPCR. **f** WT or *Gpr43*[−/−] BMDMs were transfected, treated and stimulated as in (**e**). After stimulation with polyI:C for 1 h, crude mitochondria extracts were subjected to SDD-AGE and immunoblotting as in Fig. 5h. **g** Western blot analysis of co-immunoprecipitation of MAVS with GPR43 from cell lysates of HEK293T cells transfected with a plasmid expressing V5-tagged GPR43, along with a plasmid expressing FLAG-tagged MAVS using Lipofectamine 2000. **h** BMDMs were pretreated with or without 250 μM SA for 24 h and then transfected with 200 ng polyI:C for 1 h. Fluorescence images of MAVS (red), GPR43 (green) and cell nuclei (blue) were captured with confocal microscope. Results represent two (**a–c**, **e–h**) or three independent experiments (**d**). Data in (**a–e**) are presented as mean (± SEM), two-tailed Student's *t* test (**a–d**), one-way ANOVA with Dunnett's post-hoc test (**e**). Significant values are defined by *$p < 0.05$, **$p < 0.01$, ***$p < 0.001$. Source data are provided as a Source Data file.

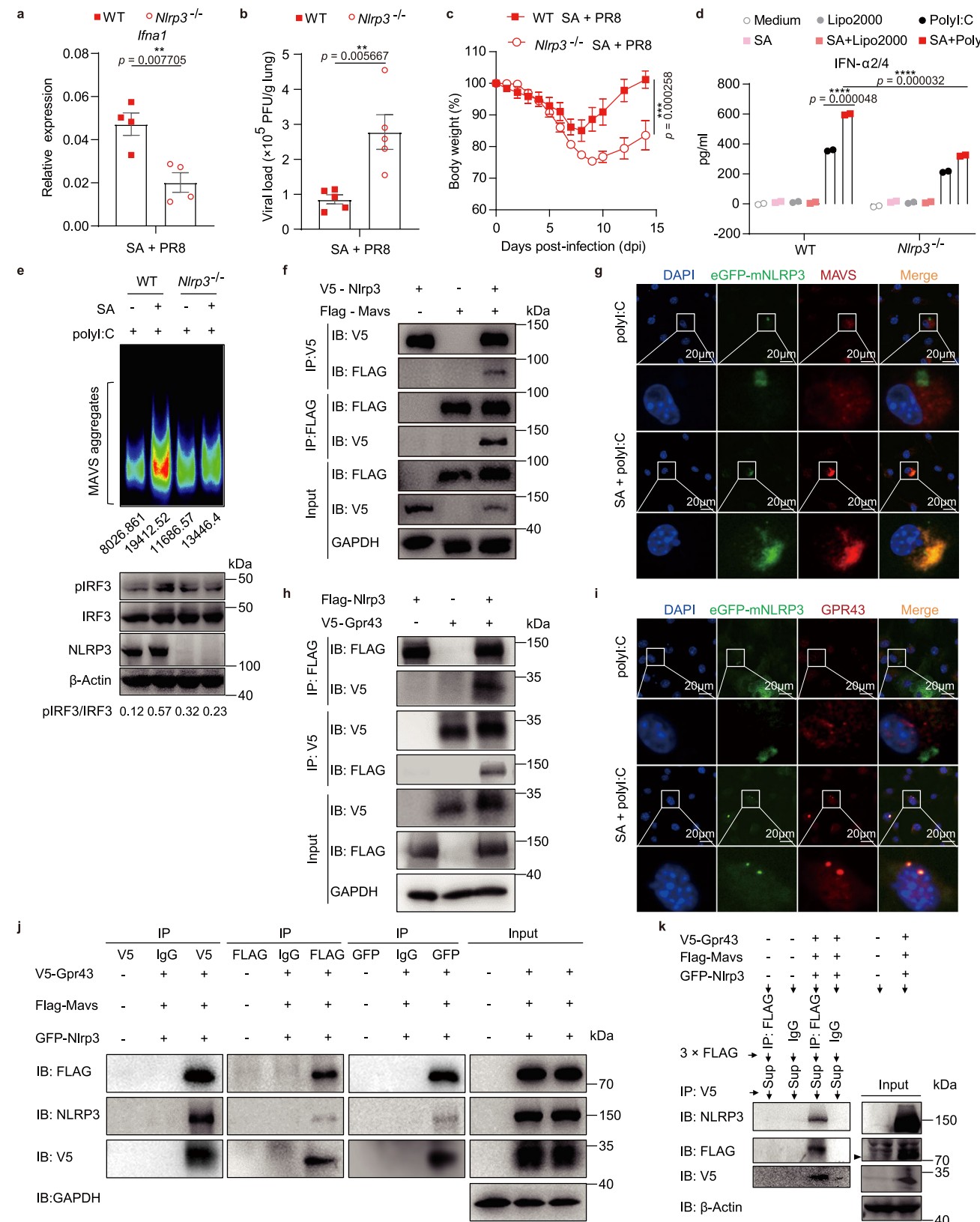

Furthermore, we found that NLRP3 participated in acetate-enhanced MAVS aggregation, as acetate enhanced MAVS aggregation and IRF3 phosphorylation in WT cells but not in *Nlrp3*⁻/⁻ cells (Fig. 7e). It has been reported that MAVS interacts with NLRP3 to facilitate NLRP3 inflammasome activation[30,31]. We confirmed that NLRP3 interacts with MAVS via Co-IP and confocal experiments

(Fig. 7f, g, Supplementary Fig. 8e), which could be a necessary step for an optimal aggregation and function of MAVS to activate IRF3 phosphorylation. Since GPR43 also interacts with MAVS (Fig. 6g, h), we investigated the possible interaction of NLRP3 with GPR43. Results of the Co-IP and confocal experiments showed that NLRP3 indeed interacted with GPR43 (Fig. 7h, i, Supplementary Fig. 8e) and

**Fig. 7 | NLRP3 participates in acetate-promoted IFN-I production via aiding MAVS aggregation.** **a–c** WT and *Nlrp3*⁻/⁻ mice were given 50 mM SA in drinking water and then intranasally infected with PR8 as depicted in Fig. 3c: relative *Ifna1* expression to *Gapdh* (n = 4, n = 4) (**a**) and viral load in lung homogenates (n = 5, n = 5) (**b**) on day 7 post-infection and weight loss (n = 10, n = 12) (**c**) were assessed. **d–e** WT and *Nlrp3*⁻/⁻ BMDMs were pretreated with 250 μM SA for 24 h and then transfected with 200 ng polyI:C for 12 h (**d**) (n = 2) or 1 h (**e**). **d** Concentrations of IFN-α2/4 released from BMDMs. **e** Crude mitochondria extracts were subjected to SDD-AGE and immunoblotted with anti-MAVS antibody, and the intensity of MAVS aggregation was analyzed with ImageJ v2.0 and marked below. The cell lysates were subjected to immunoblot analysis. The relative ratios of pIRF3 (Ser396) to total IRF3 were analyzed with ImageJ v2.0 and marked below. **f, h, j** Western blot analysis of co-immunoprecipitation of NLRP3 with MAVS or GPR43 from cell lysates of

HEK293T cells transfected with plasmids expressing V5-tagged NLRP3 and FLAG-tagged MAVS (**f**) or FLAG-tagged NLRP3 and V5-tagged GPR43 (**h**) or V5-tagged GPR43, eGFP-tagged NLRP3 and FLAG-tagged MAVS (**j**). **g, i** PRP-eGFP-mNlrp3-retrovirus infected BMDMs were pretreated with 250 μM SA for 24 h and then transfected with 200 ng polyI:C for 1 h. Fluorescence images of eGFP-mNLRP3 (green), MAVS (red) or GPR43 (red) and cell nuclei (blue) were captured with confocal microscope. **k** Western blot analysis of sequential IP in overexpressing systems. The mutual interaction among NLRP3, MAVS and GPR43 were confirmed. The black arrow indicates the specific FLAG-MAVS band. Results represent two (**a–c, g, i, k**) or three independent experiments (**d–f, j**). Data in (**a–d**) are presented as mean (± SEM), two-tailed Student's *t* test (**a–c**), one-way ANOVA with Dunnett's post-hoc test (**d**). Significant values are defined by *p < 0.05, **p < 0.01, ***p < 0.001, ****p < 0.0001. Source data are provided as a Source Data file.

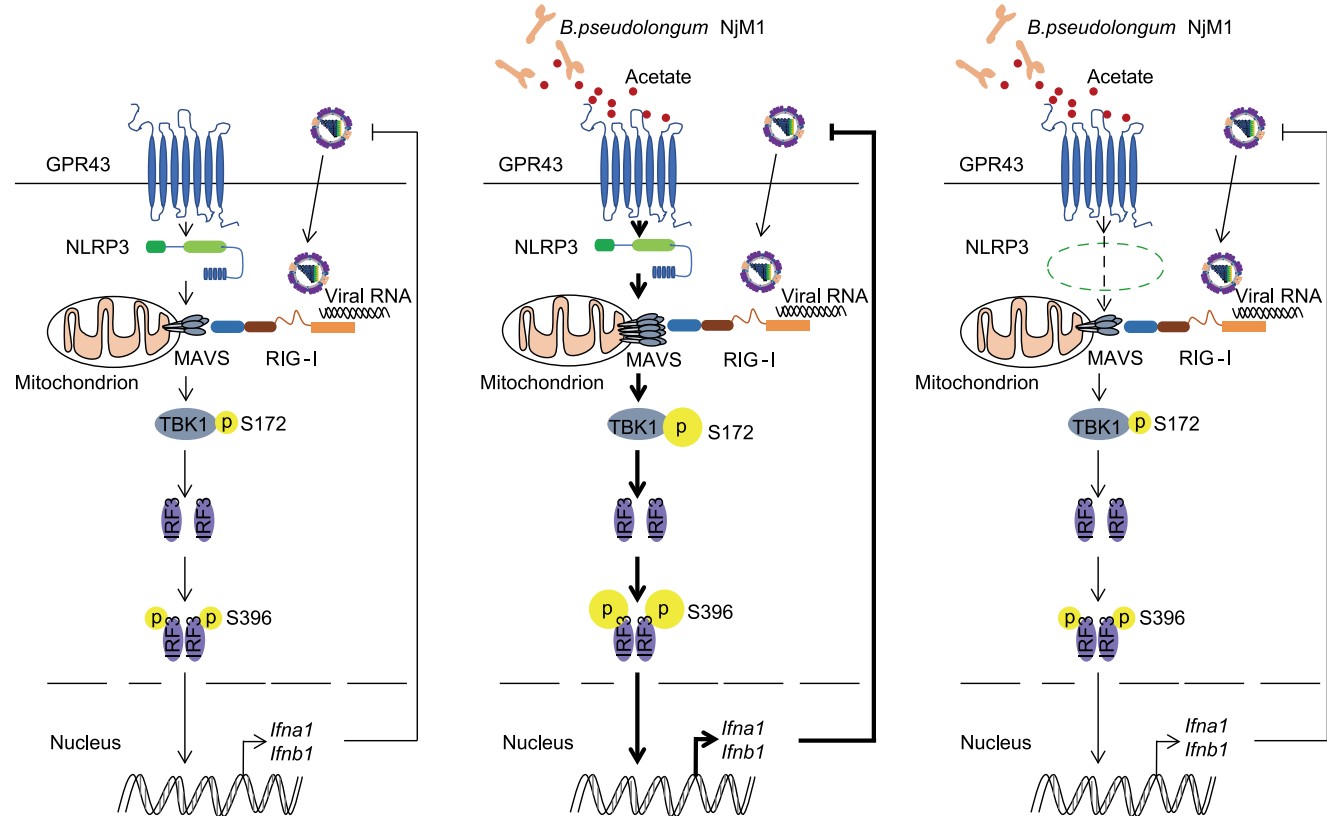

**Fig. 8 | Proposed model for the role of NLRP3 in mediating acetate-enhanced induction of IFN-I.** Recognition of influenza viral RNA by RIG-I triggers MAVS aggregation, leading to IRF3 phosphorylation (pIRF3) that initiates type I interferon (IFN-I) production to contain the virus infection (**left**). *B. pseudolongum* NjM1 produces acetate that enhances viral RNA-triggered MAVS aggregation through GPR43 and NLRP3. The enhanced MAVS aggregation promotes subsequent IRF3 activation and IFN-I production, which strongly inhibits influenza virus propagation (**middle**). NLRP3 deficiency reduces the metabolite acetate-enhanced MAVS aggregation, IRF3 phosphorylation and IFN-I production that combats the virus infection (**right**).

NLRP3 was able to form a complex with GPR43 and MAVS (Fig. 7j, k). Furthermore, we performed additional experiments to demonstrate the function of NLRP3 in the process of IFN-I generation, and found that GPR43 activation significantly increased IFN-I production, IRF3 phosphorylation and mitochondrial MAVS aggregation (Supplementary Fig. 9a–c), in which NLRP3 was required (Supplementary Fig. 9a, b, c). In addition, further results demonstrated that GPR43 activation leads to the translocation of MAVS to mitochondria, which is compromised when NLRP3 was deleted in macrophages (Supplementary Fig. 9d).

Together, these data suggested that NLRP3 interacts with GPR43, which then aids in MAVS translocation and aggregation to induce optimal IFN-I production upon IAV infection.

## Discussion

Our current work demonstrates that gut microbiota-derived acetate conferred protection against IAV infection by enhancing IFN-I production, in which NLRP3 is required to interact with the acetate receptor GPR43 to promote MAVS aggregation that activates IRF3 (Fig. 8).

The present work revealed that a new bacterium strain *Bifidobacterium pseudolongum* NjM1 is enriched in the gut microbiota of *Nlrp3*⁻/⁻ mice, which protects wild-type mice against IAV infection during co-housing. It's likely that a reciprocal regulation exits between host genetic factors and gut microbiota. In the absence of *Nlrp3* gene, probably the expression of some host factors such as antimicrobial peptides have been changed, so as to impact the composition of

microbiota in the intestine. Host shaping of gut microbiota is an important, albeit complex issue to address. The mechanism through which *Nlrp3* affects gut microbiota will be dissected in the future.

*B. pseudolongum* NjM1 produces acetate, and both the bacterium and sodium acetate are protective against IAV infection. However, it is important to note that WT mice co-housed with *Nlrp3⁻/⁻* mice (Co-WT) survived 100%, and their weight loss did not exceed 10% (Fig. 1a); in contrast, WT mice gavaged with the NjM1 strain still exhibited clear weight loss, although significantly less than in the control group (Fig. 3a). These data demonstrate that there are other protective bacteria besides the acetate-producing NjM1 in the gut of Co-WT mice against IAV infection; but NjM1 is the one that can colonize WT mice during co-housing and produce acetate to exert a protective function, even though it is not the only one. Similarly, although multiple metabolites may account for the protective effect of NjM1 against IAV infection, acetate is likely to be one of the most important. An acetokinase, whose gene is located at 1517624–1518847 bp in the genome of *B. pseudolongum* NjM1, should be the key enzyme that controls the final production of acetate. Since genetic manipulation of *Bifidobacterium* is rather challenging for the time being, we may generate mutant strain to show the direct effect of acetokinase in NjM1 on acetate production, as well as the compromised effect of such mutant strain on IAV protection in future.

It was recently demonstrated that acetate protects against respiratory syncytial virus (RSV) infection through a GPR43-IFN-I response in lung epithelial cells[32]. There is also study suggesting that alveolar macrophages produce type I interferon to fight against RSV infection[33]. Our current work demonstrated that acetate combated influenza A virus infection by enhancing viral RNA-induced IFN-I production in CD11b⁺F4/80⁺ macrophages in the lung, in which GPR43 expression was upregulated by acetate treatment. Although acetate may have also affected the function of lung epithelial cells, our experiment showed that macrophages should be one of the major types of cells for controlling IAV, as depletion of macrophages rendered acetate ineffective (Fig. 4d).

Although acetate functions as a booster in IFN-I production via GPR43, the mechanism remained elusive so far[32]. Notably, our study demonstrated that GPR43 is associated with MAVS, one of the key adaptors in antiviral signaling. Engagement of GPR43 to the MAVS-TBK1-IRF3 signalosome strongly enhanced viral RNA-induced generation of type I interferon. This finding represents a key step forward in understanding the mechanism behind GPR43-mediated enhancement of IFN-I induction. Because type I interferons have broad-spectrum antiviral functions, the mechanism found in the current work may also be applicable to resistance to other virus infections such as SARS-CoV-2.

Of note, NLRP3 is also a critical element of the acetate-GPR43-MAVS signaling complex. NLRP3 expression is upregulated by acetate once polyI:C stimulation is added (Supplementary Fig. 8a), implying that acetate makes the cells faster in inducing NLRP3 expression in response to polyI:C, which potentially promoted IFN-I production quicker when cells encounter viral infections. Acetate enhancing NLRP3 and IFN-I expression is mediated by GPR43 activation-Ca2⁺ mobilization (Supplementary Fig. 7a–d), in which NF-κB (nuclear factor κB) and p38 MAPK (mitogen-activated protein kinase) also participate (Supplementary Fig. 7e–h). Acetate alone does not induce MAVS aggregation unless the cells encounter danger signals, which should be necessary for the host to avoid pathologic damage caused by excessive IFN-I. Our biochemical and cellular evidence showed that NLRP3 is associated with both GPR43 and MAVS, and the presence of NLRP3 significantly enhanced the aggregation of MAVS, which is a prerequisite for TBK1/IRF3 phosphorylation and induction of IFN-I gene expression. Although the NLRP3 inflammasome has been implicated in many diseases, and a big plethora of cellular events affect NLRP3 activation[34,35], its participation in IFN-I induction has not been found before. Thus, the current work emphasizes the important roles that NLRP3 play in many signaling events, and implies the complexity to understand this critical molecule.

In summary, our work identified a special strain *Bifidobacterium pseudolongum* NjM1 to be protective against influenza A virus infection through acetate via GPR43-MAVS-IRF3-mediated IFN-I signal, in which NLRP3 is a key component bridging GPR43 and MAVS. Such findings imply the complex and critical connection between gut-lung, microbiota-immunity axes in combating influenza virus infection, thus may help future designing of intervention strategies against respiratory viral infections.

## Methods

### Ethics statement
All animal experiments were performed in compliance with the Regulations for the Care and Use of Laboratory Animals issued by the Ministry of Science and Technology of the People's Republic of China, which enforces the ethical use of animals. The protocol was approved by Institutional Animal Care and Use Committee (IACUC) at Institut Pasteur of Shanghai, Chinese Academy of Sciences (Permit Number: P2019014).

### Virus
Influenza virus strain A/Puerto Rico/8/1934 (H1N1) was inoculated into the chorioallantoic fluid of 9-day-old specific-pathogen-free (SPF) chicken embryos, as previously described[36]. The titer corresponding to the median lethal dose (LD₅₀) was determined in adult mice by intranasal inoculation. All viral infection experiments were performed in the biosafety level 2 laboratory at Institut Pasteur of Shanghai following the standard operating protocols approved by the institutional biosafety committee.

### Mice
All mice used in this study were female mice on C57BL/6 background (6–9 weeks old). WT (wild-type), *Ifnar1⁻/⁻* and *Mavs⁻/⁻* mice were originally from the Jax lab, *Irf3⁻/⁻* mice were originally from Dr. T. Taniguchi and shared through Dr. F. Shao[37], *Gpr43⁻/⁻* mice were purchased from Cyagen Biotechnology Co., Ltd (https://www.cyagen.com/cn/zh-cn/sperm-bank-live/233079) and *Nlrp3⁻/⁻* mice had been described before[38,39]. The mice were routinely maintained in a pathogen-free animal facility at a temperature of 21 °C, relative humidity of 50%-70%, and under a constant 12 h light/dark cycle, and were given free access to a regular chow diet (Cat# P1101F-25, Shanghai SLACOM) and water throughout the study at Institut Pasteur of Shanghai. Cohousing (Co-) of WT and *Nlrp3⁻/⁻* mice were carried out at 1:1 ratio since weaning (3 weeks old) till adulthood (9 weeks old), or the mice were separated upon weaning according to genotype, denoted as singly housed (Single-). All procedures were conducted in compliance with a protocol approved by the IACUC at Institut Pasteur of Shanghai, Chinese Academy of Sciences, China.

### Bone marrow-derived macrophages (BMDM)
BMDMs were differentiated according to the method described previously[40]. Briefly, bone marrow cells were isolated by flushing femurs and tibia of 6- to 8-week-old WT, *Nlrp3⁻/⁻*, *Irf3⁻/⁻*, *Gpr43⁻/⁻* or *Mavs⁻/⁻* mice with DMEM medium (Hyclone, Cat# SH30809.01). Red blood cells were lysed with red blood cell lysis buffer (Beyotime), and bone marrow cells were cultured in DMEM medium supplemented with 30% mouse fibroblasts L929 (ATCC CCL-1) conditioned medium, 10% FBS (AusGeneX, Cat# SA500S), 100 U/mL Penicillin and 100 μg/mL Streptomycin (Gibco), as well as 50 μM β-Me (SIGMA). At day 3, fresh DMEM medium and L929 cell (ATCC CCL-1) conditioned medium was added. BMDMs were used on day 5 for siRNA transfection by electroporation or on day 6 for sodium acetate treatment plus polyI:C stimulation.

## Virus infection of mice

Mice were anesthetized with Avertin (2,2,2-tribromoethanol, SIGMA) and infected intranasally with A/Puerto Rico/8/1934 (H1N1) in 25 uL of PBS (1 LD50 for 9-week-old WT mice). Mice were either weighed and monitored daily or euthanized at various intervals for sampling. Mice were humanely euthanized when infection had progressed to a human endpoint at which the mice started to suffer. The infected mice of more than 25% weight loss were treated as death. Weight loss and survival data were analyzed using the GraphPad Prism v8.0 software unless stated otherwise.

## Depletion of commensal bacteria

Gut microbiota removal was performed as previously described[41]. Briefly, mice were treated with a cocktail of antibiotics containing ampicillin (1 g/L), metronidazole (1 g/L), neomycin (1 g/L), and vancomycin (0.5 g/L) in drinking water for 6 weeks. Stool pellets (~0.05 g/ mice) were collected in 1 mL of Ringer's de solution (for 1 L preparation, NaCl 9 g, KCl 0.4 g, $CaCl_2 \cdot 2H_2O$ 0.25 g, and L-Cystine·HCl 0.5 g, 121 °C autoclaved for 15 min) and homogenized. Different dilutions of the suspensions were plated on mGAM (HB8518) and incubated at 37 °C either in an anaerobic incubator (80% N2:10% CO2:10% H2) or normal aerobic condition for 48 h. Bacterial counts were determined by colony-forming assay and the depletion efficiency was >99.9%. All the antibiotics used here were purchased from Sangon Biotech (Shanghai) Co.

## Gut microbiota profiling

The bacterial DNA extraction from fecal samples of mice were performed as previously described[42]. A sequencing library of the V3–V4 regions of the 16S rRNA gene was constructed following the manufacturer's instructions (Part #876 15044223Rev.B; Illumina Inc., San Diego, CA, USA) with improvement as previously described[43], and sequenced on the Illumina MiSeq platform (Illumina, Inc., San Diego, CA, USA). The raw paired-end reads were processed and analyzed using the QIIME2 v2018.11. Demultiplexed sequence data was imported into QIIME2 v2018.11, adapters and primers were trimmed. ASVs from each sample were inferred by using the DADA2 pipeline for filtering, dereplication, sample inference, merging of paired-end reads and chimera identification. In the process of running the DADA2 pipeline, based on the quality profile of the data, forward and reverse reads were trimmed accordingly to ensure that the median quality score for each position is above 32. The taxonomy of all ASVs were annotated by SILVA (v132) reference database[44]. All samples were rarefied to 10,000 per sample for downstream analysis. Principal coordinate analysis (PCoA) of ASVs based on Bray−Curtis distance was performed using QIIME2 v2018.11. The amplicon sequence variants (ASVs) that were significantly altered between two groups were identified using the algorithm LDA Effect Size (LEfSe)[20].

## ¹H NMR (nuclear magnetic resonance) spectroscopy

Detection of metabolites was conducted by Shanghai Metabolome Institute (SMI)-Wuhan. Fecal samples were prepared by dissolving 50 mg of feces in 600 μL of 0.1 M deuterated $Na^+/K^+$ buffer (for 0.1 L preparation, 1.844 g $K_2HPO_4 \cdot 3H_2O$, 0.315 g $NaH_2PO_4 \cdot 2H_2O$, 0.001 g TSP (sodium 3-(trimethylsilyl) propionate-2,2,3,3-d4), 0.01 g $NaN_3$, PH7.5) and quick frozen-thawed three times with liquid nitrogen. The mixture was homogenized with a tissue lyser (90 s, 20 Hz) and centrifuged at 13,200 × g for 10 min at 4 °C. The supernatant (550 μL) was transferred to a 5 mm NMR tube. The fecal NMR spectra were acquired at 298 K under an automatic sample changer system (BACS, Bruker Biospin, Germany) using a Bruker AVIII 600 MHz spectrometer equipped with an inverse cryoprobe (operating at 600.13 MHz for ¹H). One dimensional ¹H NMR spectra were acquired for each fecal sample using NOESYGPPR1D sequence (RD-90°-$t_1$-90°-$t_m$-90°-ACQ). Water suppression was achieved with irradiation on the water peak during the recycle delay (RD) of 2 s and a mixing time (tm) of 100 ms. $t_1$ was set to 4 μs. The 90° pulse length was adjusted to 11 μs, and 32 transients were collected into 32k data points for each spectrum with a spectral width of 20 ppm. Data analysis had been described before[45].

## Sequence-guided bacteria isolation

Fresh fecal samples from *Nlrp3⁻/⁻* mice were collected and mixed in anaerobic sterile Ringer's de solution shown in Depletion of Commensal Bacteria section above in an anaerobic workstation (Don Whitley scientific Ltd, 798 Shipley, UK). The diluted suspension with Ringer's de solution was plated onto mGAM and incubated under anaerobic condition (80% N2, 10% CO2, and 10% H2) at 37 °C for 4 days. The 16S rRNA gene V3 region of each colony was obtained using the primer pairs listed in Supplementary Table 1. The obtained 16S rRNA gene V3 sequences were subjected to Denaturing Gradient Gel Electrophoresis (DGGE) and aligned with the ASV1 or ASV4 enriched in the gut of singly housed *Nlrp3⁻/⁻* mice. ASV4-associated strain was successfully isolated using a previously reported mGAM medium (Nissui 05426) and named *B. pseudolongum* NjM1. The 16S rRNA sequence of *B. pseudolongum* NjM1 was aligned in GenBank and the 16S rRNA gene sequences of representative strains in the genus *Bifidobacterium* were selected to build a phylogenetic tree by using the Neighbor-Joining method in MEGA 6.

## Whole-genome sequencing and analysis of *B. pseudolongum* NjM1

Whole-genome DNA of *B. pseudolongum* NjM1 was extracted with QIAamp BiOstic Bacteremia DNA Kit (QIAGEN) and sequenced with the PromethION platform (Nextomics Biosciences, Wuhan, China). Subreads of the sequence were assembled into a single complete chromosome using the HGAP 2.3.0 pipeline[46]. Protein-coding sequences (CDSs), tRNAs, and rRNAs were predicted and annotated using the Prokka 1.12 pipeline[47]. Functional annotation of genes was performed using the Kyoto Encyclopedia of Genes and Genomes (KEGG) database.

## Short-chain fatty acid (SCFA) profiling for *B. pseudolongum* NjM1

SCFA concentrations in *B. pseudolongum* NjM1 culture supernatant were determined using gas chromatography/mass spectrometry (GC/MS) as previously described[48]. Briefly, *B. pseudolongum* NjM1 was cultured in mGAM medium under anaerobic condition (80% N2, 10% CO2, and 10% H2) at 37 °C for 8 h, and centrifuged at 16,000 × g for 15 min at 4 °C. *B. pseudolongum* NjM1 culture supernatant was filtered through a 0.22 μm nylon filter (EMD Millipore). An aliquot (50 μL) of the supernatant was acidified by adding 25 μL 50% (v/v) sulfuric acid and vortexing for 10–15 s. The organic acids were extracted by adding 100 μL of diethyl ether and vortexing for 10–15 s and standing for 2 min. After centrifugation at 12,000 × g for 5 min at 4 °C, the supernatant containing SCFAs was measured by GC on an Agilent 6890 (Agilent Technologies, CA, USA) equipped with flame ionization, thermal conductivity detectors, capillary columns and GC ChemStation software.

## *B. pseudolongum* NjM1 inoculation in mice

*B. pseudolongum* NjM1 was cultured in mGAM medium under anaerobic condition (80% N2, 10% CO2, and 10% H2) at 37 °C for 8 h. The harvested bacterial cells were washed twice with pure mGAM and resuspended in mGAM, and added with equal volume of 50% glycerol to a density of $7.4 \times 10^{10}$ cells/mL and stored at −80 °C until 10-fold dilution and inoculation by gavage with $1.48 \times 10^9$ cells for each mouse. The culture supernatant of *B. pseudolongum* NjM1 was used to treat mice by intranasal administration.

## Mice feeding with HAMS or HAMSA diets and acetate measurements

Diets HAMS (high-amylose maize starch) (fiber (4.8% cellulose and 15% high-amylose resistant starch), 18.1% protein, 7.1% fat and 17.7 mj/kg digestible energy) and HAMSA (acetylated HAMS) (fiber (4.8% cellulose and 15% acetylated high-amylose resistant starch), 18.1% protein, 7.1% fat and 17.7 mj/kg digestible energy) were produced by XieTong Biology, NanJing, China. Mice were fed with the diets HAMS or HAMSA for 5 weeks (starting at 3 weeks of age as previously described[21], diets were refreshed three times per week) before subjected to influenza A viral infection experiments. Cecal, fecal samples and serum were processed for SCFA analysis using gas chromatography at the end of the viral infection.

## Pulmonary histopathology

For histopathologic examination, lungs were collected from mice at the indicated days, fixed through immersion in 10% buffered formalin for at least one week at 4 °C, processed, embedded in paraffin, sectioned, and then stained with hematoxylin and eosin (H&E). The peribronchial inflammation was scored according to Barends et al[49]., ranked as absent (0), minimal (1), slight (2), moderate (3), marked (4), or severe (5). Slide analysis was performed in a blinded manner.

## Plaque assay of IAV

Lungs were weighed and homogenized in 1 mL of PBS, bronchoalveolar lavage fluid (BALF) was harvested in 1 mL of PBS, and the supernatants of PR8-infected BMDMs were harvested. They were centrifuged at 400 × g for 5 min at 4 °C. The supernatants containing influenza virus were diluted serially with 0.2% BSA (bovine serum albumin) in PBS. MDCK cells (ATCC CCL-34) were grown in 48-well cell culture plates to produce a confluent monolayer, and then washed with PBS, infected with serially diluted supernatants, and incubated at 37 °C for 1 h for virus absorption. Unabsorbed virus particles were washed away with PBS, and then 0.5 mL of 1×DMEM (SIGMA, Cat# D5796) medium containing 0.2% BSA, 100 U/mL Penicillin and 100 µg/mL Streptomycin, 1.2% colloidal microcrystalline cellulose (Avicel®) and 2 µg/mL TPCK-treated trypsin, was added to each well. After incubation for 48 h at 37 °C in 5% $CO_2$ incubator, the MDCK cells were fixed with 4% paraformaldehyde (PFA), and then incubated with rabbit anti-influenza A Nucleoprotein/NP antibody (SinoBiological, Cat# 11675-T62, dilution 1:2000) followed with goat-anti-rabbit HRP. Color development was performed with True Blue™ Peroxidase Substrate (KPL), plaques in each well were counted, and then the results were averaged and multiplied by the dilution factor for calculation of viral titers.

## Quantitative real-time PCR

Total RNA was extracted from homogenized lung or BMDMs with TRIzol reagent (SIGMA) according to the manufacturer's instructions. Synthesis of cDNA was performed with a GoScript™ Reverse Transcription kit (Promega). Real time quantitative PCR was performed with the SYBR Green qPCR Master Mix (TOYOBO) on an ABI 7900 HT Fast Real-Time cycler (Applied Biosystems). The expression of target genes was normalized to expression of housekeeping gene *Gapdh* (glyceraldehyde-3-phosphate dehydrogenase). The qPCR primers used in this study were listed in Supplementary Table 1.

## Macrophage depletion

WT mice were pretreated with sodium acetate (SA) or drinking water and then intranasally administered with 100 µL of clodronate (LIPOSOMA, C28J0620) to deplete macrophages in the lung. The cell number of alveolar macrophages in BALF were analyzed by flow cytometry.

## BALF related experiments

The bronchoalveolar lavage fluid (BALF) was harvested in 1 mL of PBS on day 3 post-infection and centrifuged at 400 × g for 5 min at 4 °C.

The supernatants were collected and frozen at −80 °C for enzyme-linked immunosorbent assay (ELISA) and virus titration. The red blood cells were lysed, and remaining cells were counted with a hemocytometer and stained with fluorochrome-conjugated antibodies for flow cytometry.

## Flow cytometry

Cells from BALF or BMDMs were washed with FACS buffer (1 × PBS containing 0.5% BSA and 2 mM EDTA) and then blocked with FcR blocking antibody (anti-Mouse CD16/CD32 (clone 2.4G2, BD Pharmingen, Cat# 553142, dilution 1:100)) before surface staining with BV605 anti-Mouse CD45.2 (clone 104, BD Horizon, Cat# 563051, dilution 1:160), APC-Cy7 anti-Mouse CD11b (clone M1/70, BD Pharmingen, Cat# 561039, dilution 1:160), PE-Cy7 anti-Mouse F4/80 (clone BM8, Biolegend, Cat# 123114, dilution 1:160) and AF488 anti-GPR43 (Bioss, Cat# bs-13536R-A488, dilution 1:160). For intracellular $Ca^{2+}$ detection, BMDMs were treated with 250 µM sodium acetate (SA) (S5636, SIGMA) for 24 h, washed with PBS three times, and then incubated with 2 µM of Fluo-4 AM (S1060, Beyotime) for 45 min at 24 °C. Dead cells were excluded using 7AAD (BD Pharmingen, Cat# 559925, dilution 1:500) and data were acquired using a BD Fortessa flow cytometer with BD FACSDivaTM software v6 and then analyzed with FlowJo_v10 software.

## Enzyme-linked Immunosorbent Assay (ELISA)

Mouse IFN-α and IFN-β protein levels were measured with mouse IFN-α ELISA kit (BMS6027, Invitrogen) and mouse IFN-β ELISA kit (42400-1, PBL Assay Science) according to the manufacturers' instructions.

## Immunoblotting

The cells were incubated in lysis buffer containing 50 mM Tris-HCl (PH 7.5), 150 mM NaCl, 1% NP40 (v/v), 1.0% SDS (m/v), 1 mM NaF, 1 mM $Na_3VO_4$ and protease inhibitor cocktail at 4 °C for 30 min on a shaker, and then the cell lysates were mixed with SDS loading buffer. The protein-loading buffer mixture was boiled to denature at 100 °C for 10 min before the SDS-PAGE gel electrophoresis for protein separation. After electrophoresis, the proteins were wet-transferred to nitrocellulose membranes (Millipore), blocked with 5% BSA in 1×TBS containing 0.05% Tween-20 and then blotted for indicated proteins using the antibodies listed in Supplementary Table 2. The uncropped blot images can be found in Source Data file.

## siRNA design and transfection

SiRNAs (small interfering RNAs) targeting *Gpr43* or *Luciferase GL2* were ordered from NIBS, Beijing. Oligonucleotides for siRNAs targeting *Gpr43* or *Luciferase GL2* were listed in Supplementary Table 1. siRNAs were transfected to BMDMs by electroporation (100 pmol of siRNAs per 1 × 10^6 BMDMs) with the electroporation solution and cuvettes (Mirus Ingenio®) and a nucleofector device (Lonza) equipped with a program O-013 for 48 h before sodium acetate treatment.

## Confocal fluorescence microscopy

BMDMs were treated with or without 250 µM sodium acetate for 24 h and then transfected with 200 ng polyI:C using transfection reagent Lipofectamine 2000 for 1 h. The cells were fixed with 4% paraformaldehyde (PFA) for 15 min, blocked with 1 × PBS containing 5% goat sera and 0.3% Triton X-100 for 1 h and then sequentially incubated with rabbit anti-MAVS (Cat# 4983, rodent specific, CST) for 12 h at 4 °C, goat-anti-rabbit IgG (AF555 conjugated) for 2 h at room temperature, rabbit anti-GPR43 (AF488 conjugated) (bs-13536R-A488, Bioss) for 2 h at room temperature and DAPI for 1 min at room temperature. The fluorescence images were collected on a laser capture confocal microscope (Olympus FV1200) using separate laser excitation to avoid any cross-interference between different fluorophores. For NLRP3 colocalization with MAVS or GPR43, *Nlrp3*−/− BMDMs were infected with PRP-eGFP-mNlrp3 retrovirus and pretreated with or without

250 µM sodium acetate for 24 h and then transfected with 200 ng polyI:C for 1 h. The cells were fixed, blocked, and then incubated with rabbit anti-MAVS (Cat# 4983, rodent specific, CST) or rabbit anti-GPR43 (bs-13536R, Bioss) for 12 h at 4 °C, goat-anti-rabbit IgG (AF555 conjugated) for 2 h at room temperature and DAPI for 1 min at room temperature. The fluorescence images were collected.

## Co-immunoprecipitation assay

HEK293T cells (ATCC CRL-3216) were transfected with the plasmids using transfection reagent Lipofectamine 2000 (invitrogen). At 36 h post-transfection, the cells were lysed with lysis buffer (50 mM Tris-HCl PH7.5, 150 mM NaCl, 1% NP40 (v/v), 0.5% SDS (m/v)) supplemented with a proteinase inhibitor cocktail (Roche), 1 mM NaF and 1 mM $Na_3VO_4$ for 30 min at 4 °C. The cell lysates were centrifuged at 1000 × g for 10 min at 4 °C and 10% supernatants were used as input, and 90% supernatants were 10-fold diluted with 0.5% NP40 (50 mM Tris-HCl PH7.5, 150 mM NaCl, 0.5% NP40 (v/v)) and then incubated with either 1 µg mouse anti-FLAG (clone M2, SIGMA, Cat# F3165, dilution 1:1000) or 1 µg mouse anti-V5 (clone AMC0506, ABcolonal, Cat# AE017, dilution 1:200) or 1 µg mouse anti-GFP (clone 5G4, CST, Cat# 55494, dilution 1:50) on a roller (QILINBEIER-206) overnight at 4 °C. The Protein A/G Magnetic Beads for IP (bimake) were added to the supernatants and incubated for 1 h on the roller at 4 °C, washed three times with 0.5% NP40, and then boiled in 2×Western loading buffer for 5 min at 95 °C for immunoblotting.

## Semidenaturing detergent agarose gel electrophoresis for MAVS aggregation

Semidenaturing detergent agarose gel electrophoresis (SDD-AGE) was performed for the detection of MAVS aggregation according to a published protocol with minor modifications[25]. In brief, BMDMs were treated with or without 250 µM sodium acetate for 24 h and then transfected with 200 ng polyI:C using transfection reagent Lipofectamine 2000 for 1 h. Crude mitochondria were extracted using a mitochondria isolation kit (Cat# 89874, Thermo Scientific™) according to the manufacturer's instructions on Reagent-based Method and then resuspended in 1×sample buffer (0.5 × TBE, 10% glycerol, 2% SDS, 0.0025% bromophenol blue) and loaded onto a vertical 1.5% agarose gel (BIOWEST). After electrophoresis in the running buffer (1×TBE and 0.1% SDS) for 35 min with a constant voltage of 100 V at 4 °C, the proteins were transferred to a PVDF membrane (Millipore) for immunoblotting.

## $Ca^{2+}$ chelator experiment and inhibitor experiment

BMDMs were pretreated with BAPTA-AM (Cat# HY-100545, MCE), BAY 11-7085 (Cat# HY-13453, MCE), SB203580 (Cat# 5633, CST), PD98059 (Cat# 9900, CST), or 25 µM SP600125 (Cat# 8177, CST) for 1 h and 250 µM sodium acetate (S5636, SIGMA) for 24 h and then transfected with 200 ng polyI:C using Lipofectamine 2000. Relative *Nlrp3* and *Ifna1* expression to *Gapdh* was determined by quantitative real-time PCR.

## Statistical analysis

GraphPad Prism v8.0 software (La Jolla, CA, USA) and IBM SPSS Statistics 23 were used for data analysis. Statistically significant difference was determined by two-tailed Student's *t* test for two groups or one-way ANOVA with Tukey's post-hoc test or Dunnett's post-hoc test for three or more groups. Survival curves were compared with the Log-rank (Mantel–Cox) test. $p < 0.05$ was considered a statistically significant.

## Reporting summary

Further information on research design is available in the Nature Portfolio Reporting Summary linked to this article.

## Data availability

The authors declare that the data supporting the findings of this study are available within the article, its supplementary information files or Source Data file. The 16 S rRNA gene sequences of *Bifidobacterium* strain NjM1 has been deposited in the Nucelotide database under accession code MW736893, MW736894, MW736895 and MW736897. The whole-genome sequence of *Bifidobacterium* strain NjM1 has been deposited in the Genome database under accession code PRJNA714197 (https://www.ncbi.nlm.nih.gov/nuccore/CP071805.1?report=fasta). The raw Illumina sequence data generated in this study have been deposited in the sequence read archive (SRA) database under accession code SRP310513. Source data are provided with this paper.

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

## Acknowledgements

The authors thank Dr. Shi Yu for help with SDD-AGE experiment, and Dr. Song Huang from NIBS for support on siRNA synthesis. G.M. is supported by grants from National Natural Science Foundation of China (81830049, 92269202), Ministry of Science and Technology of the People's Republic of China (2022YFC2303200), Strategic Priority Research Program of the Chinese Academy of Sciences (XDB29030303), International Partnership Program of the Chinese Academy of Sciences (153831KYSB20190008), the Shanghai Municipal Science and Technology Major Project (2019SHZDZX02), Research Leader Program (20XD1403900), Innovation Capacity Building Project of Jiangsu province (BM2020019). J.N. is supported by grants from Ministry of Science and Technology of the People's Republic of China (2022YFC2304700). A.L. is supported by grants from Ministry of Science and Technology of the People's Republic of China (2018YFA0507300). J.L. is supported by grants from National Natural Science Foundation of China (32000077).

## Author contributions

G.M. conceived the project; J.N. conducted most of the experiments; M.C., X.Y., J.L., Y.Y., Q.G., and A.L. helped with experiments; D.Z. and X.Q. provided critical reagents; J.N., C.Z., L.Z., and G.M. analyzed the data and wrote the manuscript; G.M. supervised the study.

## Competing interests

The authors declare no competing interests.
