## [Peer Review File · Nature Communications]

Microbiota-derived acetate enhances host antiviral response via NLRP3Editorial Note: This manuscript has been previously reviewed at another journal that is not operating a transparent peer review scheme. This document only contains reviewer comments and rebuttal letters for versions considered at *Nature Communications*.

REVIEWER COMMENTS

Reviewer #1 (Remarks to the Author):

The manuscript addresses an important and topical subject. It is improved from earlier versions and worthy of publication in Nature Comm.

Reviewer #2 (Remarks to the Author):

The authors performed sequential IP as suggested by me. Although the data quality is not good, it is acceptable to be used as supporting data for the complex formation of these three proteins.

As for the reply to my question 5, based on the data shown in Figure R13a, I disagree with the authors' comment since SA did not or did not significantly affect the expression of NLRP3. Figure 6f showed that acetate-induced MAVS aggregation only appeared in SA+poly I:C but not SA alone treated cells. Thus, the basal level of NLRP3 is unlikely to be sufficient to support GPR43-mediated MAVS activation and the induction of NLRP3 could be pre-required for the GPR43-enhanced MAVS aggregation. The authors need to make a simple discussion on this issue.

Reviewer #3 (Remarks to the Author):

This paper is of potential interest to relevant field, but lacks some important points, as other reviewers mentioned especially with that of the reviewer 3. Concerns to be addressed are listed below;

1. It has already been reported that GRP43 is involved in type I IFN response during RSV infection, mechanism of which is not known yet. (2019)10:3273 | <https://doi.org/10.1038/s41467-019-11152-6>

2. On the other hand, inconsistent results was reported in *Experimental & Molecular Medicine* ((2019) 51:83). This paper showed GRP43-mediated inhibition of NLRP3.

Under those circumstances, the authors need to clarify molecular mechanism at least on following two points.

3. Is GRP43 activated directly and how is this event leading to type I IFN induction.

4. How does GRP-43 controls Nlrp3 at molecular level? What is the difference between this paper and the other previous papers?

It also is of our interest whether and/or how does GRP43 interact with and/or modulate NLRP3-mediated IL1b production other that type I IFN.

Reviewer #1 (30th Sep 2022):

The manuscript addresses an important and topical subject. It is improved from earlier versions and worthy of publication in Nature Comm.

Reply:

We are very grateful for your recognition of our work.

Reviewer #2 (30th Sep 2022):

The authors performed sequential IP as suggested by me. Although the data quality is not good, it is acceptable to be used as supporting data for the complex formation of these three proteins.

As for the reply to my question 5, based on the data shown in Figure R13a, I disagree with the authors' comment since SA did not or did not significantly affect the expression of NLRP3. Figure 6f showed that acetate-induced MAVS aggregation only appeared in SA+poly I:C but not SA alone treated cells. Thus, the basal level of NLRP3 is unlikely to be sufficient to support GPR43-mediated MAVS activation and the induction of NLRP3 could be pre-required for the GPR43-enhanced MAVS aggregation. The authors need to make a simple discussion on this issue.

Reply:

Indeed, in the absence of polyI:C stimulation, SA alone did not clearly affect the expression of NLRP3 in mRNA and protein levels (Figure R13a (see also Supplementary Fig. 8a), R13b (left)). However, NLRP3 expression was upregulated by SA pretreatment once polyI:C stimulation was added (Figure R13a (see also Supplementary Fig. 8a), R13b (right)). We have indicated the significance and ratios of NLRP3 to β -Actin in the revised version in Figure R13. These data imply that SA pretreatment made the cells faster in upregulating NLRP3 expression in response to polyI:C, which potentially promoted IFN-I production quicker when cells encounter viral infections.

Indeed, acetate-induced MAVS aggregation only appeared in SA+poly I:C-, but not SA alone-treated cells. The fact that SA enhances MAVS aggregation only when cells encounter the danger signal such as viral RNA or polyI:C may be eventually a need for self-protection against viral infections. It might be rather harmful to the host if SA alone could induce MAVS aggregation and IFN-I production, because the microbiota continually supplies SA from gut by fermentation to the immune cells in the remote organs including the lung, excessive INF-I may cause pathologic damage. The fact that NLRP3 expression was upregulated by SA pretreatment to aid IFN-I production only when cells

encounter polyI:C stimulation indicated that NLRP3 in this case worked as an adaptor in GPR43-mediated MAVS activation. We do not argue that the basal NLRP3 in the absence of poly I:C was sufficient to support the signaling, but the induced NLRP3 in the presence of both SA and poly I:C promoted more MAVS aggregation.

The above discussion has also been incorporated in the revised version of our manuscript, line 411-414 and 417-419: “NLRP3 expression is upregulated by acetate once polyI:C stimulation is added, implying that acetate makes the cells faster in inducing NLRP3 expression in response to polyI:C, which potentially promoted IFN-I production quicker when cells encounter viral infections. Acetate alone doesn’t induce MAVS aggregation unless the cells encounter danger signals, which should be necessary for the host to avoid pathologic damage caused by excessive IFN-I.”

Figure R13

Figure R13. GPR43 activation by SA upregulates NLRP3 expression. a-b, WT and *Nlrp3*^{-/-} BMDMs were pretreated with or without 250 μM sodium acetate (SA) for 24 h and then transfected with 200 ng polyI:C. *Nlrp3* mRNA expression was determined by qPCR (a) and the cell lysates were subjected to immunoblot analysis with indicated antibodies (b). The ratios of NLRP3 to β-Actin were marked below the gels. Significant values are defined by **P* < 0.05.

Reviewer #3 (30th Sep 2022):

This paper is of potential interest to relevant field, but lacks some important points, as other reviewers mentioned especially with that of the reviewer 3. Concerns to be addressed are listed below;

*1. It has already been reported that GRP43 is involved in type I IFN response during RSV infection, mechanism of which is not known yet. (2019)10:3273
| <https://doi.org/10.1038/s41467-019-11152-6>*

*2. On the other hand, inconsistent results was reported in Experimental & Molecular Medicine ((2019) 51:83).
This paper showed GRP43-mediated inhibition of NLRP3.*

Under those circumstances, the authors need to clarify molecular mechanism at least on following two points.

3. Is GRP43 activated directly and how is this event leading to type I IFN induction.

Is GRP43 activated directly?

Reply:

Emerging evidence showed that acetate is the most potent activators of GPR43 and the activation of GPR43 subsequently influences Ca²⁺ mobilization^{6,7} (ref.6. Xu, M. D. et al. Acetate attenuates inflammasome activation through GPR43-mediated Ca²⁺-dependent NLRP3 ubiquitination. *Exp. Mol. Med.* 51 (2019). ref.7. Thorburn, A. N., Macia, L. & Mackay, C. R. Diet, Metabolites, and "Western-Lifestyle" Inflammatory Diseases. *Immunity* 40, 833-842 (2014).). Then we detected whether SA (Sodium Acetate) pretreatment causes Ca²⁺ mobilization so as to answer whether GPR43 has been activated directly in our experimental settings. As shown in Figure R20 (added to Supplementary Fig. 7a, b), SA pretreatment induced an increase in intracellular Ca²⁺ in WT but not *Gpr43*^{-/-} BMDMs. This demonstrated that GPR43 is activated directly by SA. We have interpreted the data (line 292-294) and the methodology (line 642-645) in the revised manuscript.

Figure R20

Figure R20. Activation of GPR43 by sodium acetate subsequently influences Ca²⁺ mobilization. **a-b**, WT or *Gpr43*^{-/-} BMDMs were treated with 250 μ M sodium acetate (SA) or medium for 24 h, washed with PBS three times, and then incubated with 2 μ M of Fluo-4 AM (a Ca²⁺ fluorescence probe) diluted in PBS for 45 min at 24°C. The percentage of Ca²⁺ positive cells (**a**, **b**) and mean fluorescence intensity (MFI) of Ca²⁺ (**b**) was analyzed by flowcytometry. Significant values are defined by * $P < 0.05$, ** $P < 0.01$.

How is this event leading to type I IFN induction?

Reply:

We found that GPR43 activation by SA upregulates NLRP3 expression in mRNA and protein levels (Figure R13a (see also Supplementary Fig. 8a), R13b, Supplementary Fig. 9d (Total lysates)) and subsequently makes NLRP3 help MAVS (mitochondrial antiviral signaling (MAVS) protein translocate to the mitochondria (Supplementary Fig. 9d (Raw Mitochondria)) and aggregate (Supplementary Fig. 9c) upon viral infection. It has been reported that MAVS aggregation activates the transcription factor IRF3 to induce type I interferons⁸ (ref. 8. Hou, F. et al. MAVS forms functional prion-like aggregates to activate and propagate antiviral innate immune response. *Cell* 146, 448-461 (2011)). In our experimental settings, NLRP3 mediated MAVS aggregation indeed initiates IRF3 phosphorylation (pIRF3) (Supplementary Fig. 9b) that induces IFN-I production

(Supplementary Fig. 9a).

Figure R13

Figure R13. GPR43 activation by SA upregulates NLRP3 expression. a-b, WT and *Nlrp3*^{-/-} BMDMs were pretreated with or without 250 μM sodium acetate (SA) for 24 h and then transfected with 200 ng polyI:C. *Nlrp3* mRNA expression was determined by qPCR (a) and the cell lysates were subjected to immunoblot analysis with indicated antibodies (b). The ratios of NLRP3 to β-Actin were marked below the gels. Significant values are defined by **P* < 0.05.

Supplementary Fig. 9 NLRP3 mediates translocation of MAVS triggered by GPR43 activation to mitochondria. **a**, WT and *Nlrp3*^{-/-} BMDMs were pretreated with or without 20 μ M 4-CMTB for 24 h, and then transfected with 200 ng polyI:C using Lipofectamine for 12 h. Concentrations of IFN- α 2/4 released from BMDMs were determined by ELISA. one-way ANOVA with Dunnett's post-hoc test. Significant values are defined by *** $P < 0.01$. **b**, WT or *Nlrp3*^{-/-} BMDMs were pretreated with or without 20 μ M 4-CMTB for 24 h and then transfected with 200 ng polyI:C for 1 h or 2 h. The cell lysates were subjected to immunoblot analysis with indicated antibodies. **c**, WT or *Nlrp3*^{-/-} BMDMs were pretreated and transfected for 1 h as shown in (b), crude mitochondria extracts were prepared from the BMDMs and then subjected to SDD-AGE and immunoblotted with rabbit anti-MAVS antibody. **d**, WT, *Nlrp3*^{-/-} or *Gpr43*^{-/-} BMDMs were pretreated with or without 20 μ M 4-CMTB for 24 h, crude mitochondria extracts and cell lysates were subjected to immunoblot analysis with indicated antibodies. Results represent n=2 independent experiments (a-d). Significant values are defined by * $P < 0.05$, ** $P < 0.01$, *** $P < 0.001$. Source data are provided as a Source Data file. 4-CMTB, the agonist for GPR43.

4. How does GRP-43 controls Nlrp3 at molecular level? What is the difference between this paper and the other previous papers?

How does GRP-43 controls Nlrp3 at molecular level?

Reply:

Before polyI:C stimulation, SA-mediated GPR43 activation (Figure R20, added to Supplementary Fig. 7a, b) did not significantly affect the expression of NLRP3 in mRNA and protein levels (Figure R13a (see also Supplementary Fig. 8a), R13b (left)). However, NLRP3 expression was upregulated by SA pretreatment once polyI:C stimulation was added (SA+polyI:C) (Figure R13a (see also Supplementary Fig. 8a), R13b (right)), and this was dependent on GPR43 (Figure R21a, added to Supplementary Fig. 7c). We first speculate that SA+polyI:C induced Nlrp3 expression was mediated by GPR43 activation-Ca²⁺ mobilization. To this end, we used BAPTA-AM (a Ca²⁺ chelator) and then detected SA+polyI:C induced Nlrp3 expression. BAPTA-AM significantly decreased Nlrp3 expression (Figure R22a, added to Supplementary Fig. 7d). In addition, NF- κ B inhibition (Figure R22b, added to Supplementary Fig. 7e) and MAPK P38 inhibition (Figure R22c, added to Supplementary Fig. 7f) also significantly decreased Nlrp3 expression, while MAPK ERK inhibition or MAPK JNK inhibition didn't affect *Nlrp3* expression (Figure R22d, e, added to Supplementary Fig. 7g, h). These data demonstrated that the enhanced Nlrp3 expression by SA-GPR43 activation after polyI:C stimulation was dependent on Ca²⁺ mobilization, NF- κ B and MAPK P38. We have interpreted the data (line 414-417) and the methodology (line 707-712) in the revised manuscript.

Figure R13

Figure R13. GPR43 activation by SA upregulates NLRP3 expression. **a-b**, WT and *Nlrp3*^{-/-} BMDMs were pretreated with or without 250 μM sodium acetate (SA) for 24 h and then transfected with 200 ng polyI:C. *Nlrp3* mRNA expression was determined by qPCR (**a**) and the cell lysates were subjected to immunoblot analysis with indicated antibodies (**b**). The ratios of NLRP3 to β-Actin were marked below the gels. Significant values are defined by **P* < 0.05.

Figure R21

Figure R21, a-b, WT and *Gpr43*^{-/-} BMDMs were pretreated with 250 μM SA for 24 h and then transfected with 200 ng polyI:C using Lipofectamine 2000. Relative *Nlrp3* and *Ifna* expression to *Gapdh* was determined by quantitative real-time PCR. Significant values are defined by **P* < 0.05.

Figure R22

Figure R22, a-e, WT BMDMs were pretreated with 10 μ M BAPTA-AM (a Ca^{2+} chelator), 10 μ M BAY 11-7085 (a NF- κ B inhibitor), 10 μ M SB203580 (a MAPK P38 inhibitor), 20 μ M PD98059 (a MAPK ERK inhibitor), or 25 μ M SP600125 (a MAPK JNK inhibitor) for 1 h and 250 μ M SA for 24 h and then transfected with 200 ng polyI:C using Lipofectamine 2000. Relative *Nlrp3* expression to *Gapdh* was determined by quantitative real-time PCR. Significant values are defined by * $P < 0.05$, ** $P < 0.01$, *** $P < 0.001$.

What is the difference between this paper and the other previous papers?

Reply:

Our current work demonstrates that gut microbiota-derived acetate confers protection against influenza virus A (IAV) infection by enhancing IFN-I production, in which NLRP3 is required to interact with the acetate receptor GPR43 to promote MAVS aggregation that activates IRF3 (Fig. 8). Our data reveal that NLRP3 is a critical element of the acetate-GPR43-MAVS signaling and suggest that the acetate-GPR43-NLRP3-MAVS-IFN-I signaling axis is a potential therapeutic target against respiratory viral infections.

Fig. 8

Fig. 8 Proposed model for the role of NLRP3 in mediating acetate-enhanced induction of IFN-I. Recognition of influenza viral RNA by RIG-I triggers MAVS aggregation, leading to IRF3 phosphorylation (pIRF3) that initiates type I interferon (IFN-I) production to contain the virus infection (**left**). *B. pseudolongum* NjM1 produces acetate that enhances viral RNA-triggered MAVS aggregation through GPR43 and NLRP3. The enhanced MAVS aggregation promotes subsequent IRF3 activation and IFN-I production, which strongly inhibits influenza virus propagation (**middle**). NLRP3 deficiency reduces the metabolite acetate-enhanced MAVS aggregation, IRF3 phosphorylation and IFN-I production that combats the virus infection (**right**).

In our current paper, acetate (250 μM) enhances viral RNA-induced NLRP3 expression and IFN-I production in BMDMs through GPR43-NLRP3-MAVS-pIRF3-IFN-I axis to control influenza A virus (IAV) replication (see the table below).

One of the previous papers⁴ (ref. 4. Antunes, K. H. et al. Microbiota-derived acetate protects against respiratory syncytial virus infection through a GPR43-type I interferon response. *Nat Commun* 10, 3273 (2019).) showed that acetate (260 μM) enhances respiratory syncytial virus (RSV)-induced IFN-I production in human pulmonary epithelial cells (A549 and MRC-5) through GPR43-mediated NF- κB p65 translocation to nucleus to contain respiratory syncytial virus (RSV) replication (see the table below).

The other previous paper⁶ (ref.6. Xu, M. D. et al. Acetate attenuates inflammasome activation through GPR43-mediated Ca²⁺-dependent NLRP3 ubiquitination. *Exp. Mol. Med.* 51 (2019).) demonstrated that acetate (20 mM~60 mM) promoted NLRP3 degradation in mouse peritoneal macrophage (PM) and BMDMs and subsequently decreased IL-1 β production via GPR43-mediated Ca²⁺ mobilization decrease and cAMP level change to protect against NLRP3 inflammasome-dependent peritonitis and LPS-induced endotoxemia (see the table below).

This demonstrates that acetate functions differently when it acts at different concentration (from μ M to mM) on different cell types.

The difference between this paper and the other previous papers is listed below.

Paper Titles	in vivo Model	Cell types	The concentration of acetate used	Proposed Mechanisms	NLRP3 expression	IFN-I production
NLRP3 mediates acetate-enhanced induction of IFN-I to alleviate influenza A virus infection (This paper)	Influenza A virus infection	Mouse bone-marrow derived macrophages (BMDMs)	250 μ M (0.25 mM)	Acetate \rightarrow GPR43 \rightarrow NLRP3 \rightarrow MAVS \rightarrow pIRF3 \rightarrow IFN-I	polyI:C induced NLRP3 expression was upregulated by acetate (250 μ M)	Acetate enhances IFN-I production

Microbiota-derived acetate protects against respiratory syncytial virus infection through a GPR43-type 1 interferon response (One previous paper)

Human pulmonary epithelial cells (A549 and MRC-5)	Respiratory syncytial virus (RSV) infection	260 μ M (0.26 mM)	Acetate \rightarrow GPR43 \rightarrow NF- κ B p65 translocation to nucleus \rightarrow IFN-I	Not detected	Acetate enhances IFN-I production
---	---	-----------------------	---	--------------	-----------------------------------

Acetate attenuates inflammatory some activation through GPR43-mediated Ca²⁺-dependent NLRP3 ubiquitination (The other previous paper)

Mouse peritoneal macrophage (PM), Mouse bone-marrow derived macrophages (BMDMs)	NLRP3 inflammasome-dependent peritonitis and LPS-induced endotoxemia	20-60 mM	Acetate \rightarrow GPR43 \rightarrow Ca ²⁺ /cAMP \downarrow \rightarrow NLRP3 ubiquitination and degradation	NLRP3 protein was degraded by acetate (20-60 mM)	Not detected
---	--	----------	--	--	--------------

It also is of our interest whether and/or how does GPR43 interact with and/or modulate NLRP3-mediated IL1 β production other than type I IFN.

Reply:

Our data demonstrated that acetate-mediated suppression of IAV infection *in vivo* was fulfilled through type I interferon signaling (line207-line219), which ruled out the possibility that IL-1 β involved in acetate protection against IAV infection. Accordingly, acetate failed to enhance IL-1 β production *in vivo* (Figure R23b; a, recitation of Fig. 6b), and acetate alone or acetate plus Poly I:C did not induce IL-1 β release from macrophages *in vitro* (Figure R3b, also added to Supplementary Fig. 8a). In addition, acetate did not change IL-1 β release from macrophages stimulated with LPS plus ATP (Figure R19). These results showed that SA-GPR43 activation don't modulate NLRP3-mediated IL-1 β production *in vivo* and *in vitro* in our experimental settings. A paper from Xu et al. showed that acetate at the concentration of 20mM - 60 mM, especially 60 mM, strongly inhibited IL-1 β release from BMDMs treated with LPS + Nigericin or LPS+ATP⁶ (ref.6. Xu, M. D. et al. Acetate attenuates inflammasome activation through GPR43-mediated Ca²⁺-dependent NLRP3 ubiquitination. *Exp. Mol. Med.* 51 (2019).). This indicated that 20 mM – 60 mM but not 250 μ M (0.25 mM) of acetate suppressed NLRP3-mediated IL-1 β production.

Figure R23

Figure R23, Concentrations of IFN- α 2/4 and IL-1 β in the BALF from infected WT mice were determined by ELISA on day 3 post infection. *, p<0.05.

Figure R3

Figure R3, a-b, WT and *Nlrp3*^{-/-} BMDMs were pretreated with or without 250 μM sodium acetate (SA) for 24 h and then transfected with 200 ng polyI:C using Lipofectamine for 12 h. Concentrations of IFN-α2/4 and IL-1β released from BMDMs were determined by ELISA.***, p<0.001.

Figure R19

Figure R19, WT and *Nlrp3*^{-/-} BMDMs were pretreated with or without 250 μM sodium acetate (SA) for 24 h and then primed with LPS (500 ng/ml) for 3 h followed by stimulation with ATP (4mM) for 30 min. Concentration of IL-1β released from BMDMs was determined by ELISA.

The data from Xu et al. is displayed below.

a-b ELISA of IL-1β in BMDMs treated with LPS + nigericin or LPS+ ATP at different

doses (20, 30, 35, 40, and 60 mM) (n =3).

References cited in this response letter

1. Marino, E. et al. Gut microbial metabolites limit the frequency of autoimmune T cells and protect against type 1 diabetes. *Nat. Immunol.* **18**, 552–562 (2017).
2. Macia, L. et al. Metabolite-sensing receptors GPR43 and GPR109A facilitate dietary fibre-induced gut homeostasis through regulation of the inflammasome. *Nat Commun* **6**, 6734 (2015).
3. Furuhashi, T., Sugitate, K., Nakai, T., Jikumaru, Y. & Ishihara, G. Rapid profiling method for mammalian feces short chain fatty acids by GC-MS. *Anal. Biochem.* **543**, 51–54 (2018).
4. Antunes, K. H. et al. Microbiota-derived acetate protects against respiratory syncytial virus infection through a GPR43-type 1 interferon response. *Nat Commun* **10**, 3273 (2019).
5. Barends, M. et al. Timing of infection and prior immunization with respiratory syncytial virus (RSV) in RSV-enhanced allergic inflammation. *J. Infect. Dis.* **189**, 1866–1872 (2004).
6. Xu, M. D. et al. Acetate attenuates inflammasome activation through GPR43-mediated Ca²⁺-dependent NLRP3 ubiquitination. *Exp. Mol. Med.* **51** (2019).
7. Thorburn, A. N., Macia, L. & Mackay, C. R. Diet, Metabolites, and “Western-Lifestyle” Inflammatory Diseases. *Immunity* **40**, 833–842 (2014).
8. Hou, F. et al. MAVS forms functional prion-like aggregates to activate and propagate antiviral innate immune response. *Cell* **146**, 448–461 (2011).

REVIEWERS' COMMENTS

Reviewer #2 (Remarks to the Author):

I have no further question.

Reviewer #2 (23rd Dec 2022)

I have no further question.

Reply:

We are very grateful for your recognition of our work.